# Survey of Cooperative Advanced Driver Assistance Systems: From a Holistic and Systemic Vision

**DOI:** 10.3390/s22083040

**Published:** 2022-04-15

**Authors:** Juan Felipe González-Saavedra, Miguel Figueroa, Sandra Céspedes, Samuel Montejo-Sánchez

**Affiliations:** 1Department of Electrical Engineering, Universidad de Concepción, Concepción 4070386, Chile; juanfegonzalez@udec.cl (J.F.G.-S.); mifiguer@udec.cl (M.F.); 2Department of Computer Science and Software Engineering, Concordia University, Montreal, QC H3G 1M8, Canada; sandra.cespedes@concordia.ca; 3Programa Institucional de Fomento a la Investigación, Desarrollo e Innovación, Universidad Tecnológica Metropolitana, Santiago 8940577, Chile

**Keywords:** cooperative advanced driver assistance systems, road safety system, holistic, systemic, Internet of vehicles, natural perception, assisted perception, human–machine interface

## Abstract

The design of cooperative advanced driver assistance systems (C-ADAS) involves a holistic and systemic vision that considers the bidirectional interaction among three main elements: the driver, the vehicle, and the surrounding environment. The evolution of these systems reflects this need. In this work, we present a survey of C-ADAS and describe a conceptual architecture that includes the driver, vehicle, and environment and their bidirectional interactions. We address the remote operation of this C-ADAS based on the Internet of vehicles (IoV) paradigm, as well as the involved enabling technologies. We describe the state of the art and the research challenges present in the development of C-ADAS. Finally, to quantify the performance of C-ADAS, we describe the principal evaluation mechanisms and performance metrics employed in these systems.

## 1. Introduction

The history of research on assistance systems for driving safety is almost as old as the rise of the automobile industry itself. One of these first systems was implemented as standard equipment by Volvo in 1959, when it began to install the seat belt with a three-point system in its vehicles, which was patented in 1962 [1] and later became the universal standard used by commercial vehicles today. Different road safety mechanisms, also known as advanced driver assistance systems (ADAS), have been incorporated over the years in the development of the automotive industry to provide a higher level of road safety to traditional transport systems. These mechanisms constitute technological implementations of concepts that the Electronic and Telecommunications Systems Institute (ETSI) has subsequently defined as road safety applications, which can be grouped into primary road safety applications (equivalent to the so-called active road safety systems) and secondary and tertiary road safety applications (equivalent to so-called passive road safety systems) [2]. The former focuses on collision avoidance as its primary mission, while the latter aims to lessen the severity of potential injuries to vehicle passengers or vulnerable road users (VRU) after the collision. Specific bumpers designed to keep pedestrians safe, airbags, and seat belts are some of the well-known examples of passive safety mechanisms. Others such as the anti-lock braking system (ABS), lane change assistance (LCA), or forward collision warning (FCW) are examples of active safety systems.

Thanks to the technological development achieved in the electronics area, the automotive industry evolved from traditional mechanical control to electronic control of the main internal functions of the vehicle, fundamentally related to acceleration, braking, gear change, suspension, and ignition/alternator integration [3,4]. The principal elements of this evolution are the electronic control units (ECU) and the management of the information flow between ECUs, for which bus communication protocols, such as the controller area network (CAN) bus [5,6] or the local interconnect network (LIN), are employed. These developments allowed the implementation of safety systems such as the ABS, the electronic stability control (ESC), or electronically assisted steering (EAS). The introduction of on-board sensors such as radars, ultrasonic, cameras, light detection and ranging (LIDAR), the global positioning system (GPS), and the inertial measurement unit (IMU), enables the development of driving assistance systems more oriented to the sensing and interaction of the vehicle with the surrounding environment to help the driver in the driving process [7]. Initially, these safety systems were oriented to the longitudinal control of the movement of the vehicle, such is the case of the adaptive cruise control (ACC) system, FCW, and autonomous emergency braking (AEB) [8]. Later, lateral control-oriented systems emerged, such as the lane departure warning (LDW) [9], blind-spot warning (BSW) [10], lane-keep assistance (LKA) [11], or LCA [12]. Although these sensor technologies represented a benefit for the surrounding awareness knowledge of the vehicle in terms of road safety, their short-range and line-of-sight obstruction problems limit their ability to sense the environment. A new generation of ADAS, named cooperative advanced driver assistance systems (C-ADAS) [13,14], has started to be developed, which also incorporates wireless communication devices. The main distinguishing factor between the C-ADAS and ADAS is the visibility of the system. C-ADAS can have knowledge of the traffic situation miles ahead, while conventional systems have knowledge of a few meters around the vehicle. The extended visibility is possible not only due to vehicle-to-vehicle (V2V) communication but also to vehicle-to-infrastructure (V2I) and vehicle-to-everything (V2X) communication [15].

Road safety is an area in which internal and external elements converge and interact. Figure 1 shows the main elements involved in a road safety system (RSS), the environment, the driver, and the vehicle. The limitations in the design of ADAS by not considering the presence of all these elements, as well as the interaction among them, degrades the level of road safety and diminishes the effectiveness of these systems.

An important aspect for the proper functioning of C-ADAS is the performance of the vehicular communication networks, which must be able to guarantee the timely exchange of sensitive information for road safety. Vehicular ad hoc networks (VANETs) [16] are powered by vehicular communications and further provide an enhancement in driving experience by improving security, infotainment, and robustness. Many researchers have contributed and explored this concept, but due to many security and privacy-related issues, the implementation stage has not matured enough. Only in the last few years has there been a growing commercial interest from the automotive industry in the area of vehicular communications. A considerable volume of research in this field has led to a move from conventional VANETs to the Internet of vehicles (IoV) [17]. The fusion between the traditional concepts of VANETs and the most recent concepts linked to the Internet of Things (IoT) has promoted a very tempting research area today. The IoV concept materializes this integration process between VANETs, IoT, and mobile computing. In this sense, the so-called intelligent vehicles are defined, which are equipped with Internet connection devices to collaborate with each other through data exchange and can even interact with other road users, such as pedestrians or roadside units, through the exchange of information about the road environment. Under this concept, a traffic management system can be established based on communication and cooperation between the vehicles, the road infrastructure, and the rest of the actors present in the road environment. On-board sensors and processors with remote connection capacity are required in each element of the network, where the information exchange must be managed efficiently to guarantee that the system is useful and capable of providing a road safety benefit.

In this work, we present a review of the previous works that have addressed the interactions between the principal elements of the RSS (i.e., driver, environment, and vehicle) and the challenges that currently persist in each area. Then, we present a proposal for the general vision of the C-ADAS design. Note that, unlike the previous studies, in this work we intend to describe a holistic and systemic architecture while highlighting the main works that have already been carried out in each type of interaction. Besides, we describe the elements that we consider fundamental for proposing a C-ADAS design that seeks to bring together all the technological advantages from the point of view of algorithms, communications, and sensing capabilities. The main contributions of this work are: (i) To propose a novel C-ADAS architecture with a holistic and systemic approach that considers the main elements of the RSS (driver, environment, and vehicle) and their interactions. To the best of our knowledge, it is the only survey that addresses the interactions between each of these elements from a bidirectional approach, considering the relevance of information exchange, highlighting cooperative solutions and the need for intelligent techniques in decision making; (ii) To present a study on the main evaluation metrics and mechanisms used for the validation of C-ADAS, which has been little addressed in the previously published surveys.

The rest of this paper is organized as follows: In Section 2, we describe the main characteristics and limitations of previous surveys on ADAS and the current challenges in this subject, while in Section 3, we propose a novel C-ADAS architecture from a holistic and systemic perspective. Section 4, Section 5 and Section 6 review the details associated with the interactions of the three principal elements (driver, environment, and vehicle) of the RSS, mentioning its actual state and challenges. More specifically, Section 4 presents an overview of the interactions between vehicle and environment, defined as an assisted perception module, Section 5 details the interaction between driver and vehicle, mainly described through the HMI module, and Section 6 presents the interactions between driver and environment, defined as the natural perception module. Section 7 presents the most used metrics and mechanisms in the evaluation and validation of C-ADAS systems, mentioning some of the main challenges in this area that, according to our vision, hinder the implementation of a C-ADAS system, such as the one proposed in this article. Finally, we conclude the paper in Section 8 and consider some directions for future work.

## 2. Previous ADAS Surveys and Current Challenges

There are several published surveys on ADAS in connected and automated vehicles (CAV) with different design perspectives. Despite the latent needs and challenges of the integration and bidirectional interactions of the three subsystems analyzed, to the best of our knowledge, these aspects have not been addressed in the scientific literature. However, some surveys have stood out for the rigor and depth with which they address some of these elements. In this section, we present the aspects in which each one of them stands out in order to later point out the shortcomings and discuss the current challenges in the subject.

Pathan et al. [18] provide a review of the proposed techniques to implement C-ADAS and intelligent traffic management systems, comparing pros and cons, and also look at the practically feasible features;Hasenjäger et al. [19] provide a review of the personalization for ADAS and propose a general conceptual framework of personalized ADAS and the human–machine interface (HMI), which can be expected to continuously adapt in interaction with the driver;Martinez et al. [20] provide a survey on driving-style characterization and recognition, revising several algorithms with emphasis on machine learning approaches;Xing et al. [21] present an overview of the driver intention inference, which mainly focuses on the lane change intention on highways;Bila et al. [22] provide an overview of information and communication technologies-based support and assistance services for the safety of future connected vehicles, given from the perspective of vehicle detection, road detection, lane detection, pedestrian detection, drowsiness detection, and collision avoidance;Siegel et al. [23] summarize the state of the art in connected vehicles, reviewing the architectures, enabling technologies, applications, and development areas;Wang et al. [24] focus on heterogeneous multi-sensors fusion technologies, including radar, camera, LIDAR, ultrasonic, GPS, IMU, and V2X communication, analyzing the necessity of fusion strategies because of the limitations of sensors;Martí et al. [25] propose an overview of existing and upcoming sensor technologies applied to common perception tasks for ADAS and automated driving. Specifically, they focus on artificial vision, radar, and LIDAR technologies of exteroceptive sensors applied in tasks as (i) automatic traffic-sign detection and recognition, (ii) perception of the environment, and (iii) vehicles, pedestrians, and other obstacles detection;Kaiser et al. [26] carry out an author-centric literature review to illustrate the opportunities in using smartphones to detect driver distraction. The authors have reviewed several papers and summarized their application cases, smartphone sensor data used, methods, and results;Arumugam et al. [27] survey driver behavior analysis based on the use of big data. Works related to monitoring driving patterns and fatigue, detecting drowsiness and driver distraction, are discussed;Zhang et al. [28], investigate mean take-over times from 129 studies with SAE (Society of Automotive Engineers) level two automation or higher. How quickly drivers take-over control of the vehicle in response to a critical event or a take-over request is an important question in automated driving research;Sarker et al. [29] review the principal aspects of the sensing and communications technologies, human factors, and controller aspects for information-aware CAV.

Table 1 summarizes these works and shows that none of them have described the ADAS architecture from a holistic and systemic perspective; evaluation mechanisms are hardly mentioned and only one of them has addressed the interaction between the three subsystems, but not bidirectionally in all cases.

### Current Challenges

A holistic approach of the system in the interactions between vehicle–environment, vehicle–driver, and driver–environment, which are essential for improving road safety, has not been sufficiently addressed in the literature consulted. Holistic is an adjective that indicates that something is relative or belongs to holism. Holism is a concept created by Jan Christiaan Smuts [30], which he describes as “the tendency of nature to use creative evolution to form a whole that is greater than the sum of its parts”. In general terms, holistic indicates that a system and its properties are analyzed as a whole, in a global and integrated way, through the multiple interactions that characterize them. Holism assumes that all the properties of a system cannot be determined or explained as the sum of its elements, highlighting the importance of the inter-dependence of those elements. The term systemic is used in the literature with a similar meaning to the term holistic; however, in this work when referring to a systemic perspective we focus mainly on the structural aspect of the system, in the analysis of each of the elements that compose it, and in the bidirectional study of the interactions that occur among them.

In this sense, the design of current ADAS has a variety of potential difficulties: reflecting the effects of all kinds of traffic factors on driving safety; describing the interactions between the characteristics of the driver’s behavior, the state of the vehicle, and the road environment; or providing an accurate basis for vehicle control. Existing systems that assess driving safety may not work properly if they consider only a limited number of factors and their interactions. Driving is a complex decision-making process due to the intricate relationships between the main elements (vehicle, environment, and driver) and the dynamic nature of these elements. In the driver–vehicle–road closed-loop system, the driver is a crucial component, with unique driving characteristics that vary from driver to driver or even for the same driver under different conditions or on different days.

The directionality of the interactions between vehicle, environment, and driver is very relevant. For example, consider a situation where a driver approaches an intersection, in which the traffic light just switched to red. In this scenario, the environment is issuing a notification about the state of road safety, representing a form of interaction in the direction environment -> driver (E -> D) and environment -> vehicle (E -> V). This notification can be captured correctly or not by the driver. The vehicle, through its on-board sensors, could detect the red light and then calculate the distances between the elements of the environment, e.g., the predecessor vehicles and the intersection. Besides, the surrounding situation is transmitted to other road users through messages corresponding to the V -> E interaction. On the other hand, the vehicle can sense the driver’s reaction to surrounding stimuli by monitoring the driver through on-board devices, which is a form of D -> V interaction. The system alerts the driver through the V -> D interaction if their reaction is not as expected and communicates to other road users the potential risk, as a V -> E interaction form. Additionally, the vehicle captures the driver’s reaction to the C-ADAS alert notice, such as a D -> V interaction form. Consequently, the C-ADAS takes control of the vehicle to guarantee road safety if the driver’s reaction is not as expected, which is a form of V -> E interaction. In this way, it is observed how the C-ADAS system is responsible for assisting the driver in a timely manner on road safety risks, guaranteeing the correct closure of the cycle started with the notification of the environment to the driver on the red light (the E -> D interaction), finishing with a correct response in relation to the reaction that the driver should have in this situation (D -> E interaction), preserving road safety with or without the intervention of the C-ADAS at different levels. This shows how C-ADAS can mitigate a poor reaction on the part of the driver, resulting from a wrong interpretation of the situation, or the driver’s inability to react.

The RSS consists of a driver subsystem, a vehicle subsystem, and a road environment subsystem. These three subsystems are essential to ensure the safe driving of automobiles. The “gaps” in the interactions between the three subsystems are causes of degradation in road safety. A fundamental element to fill the gaps in these interactions is the design of ADAS from a holistic approach that considers the presence and interaction between these different elements as a single system. Recently, there have been situations in which the challenges faced by the automation of vehicle systems are emphasized, demonstrating how the non-consideration of any of these elements can produce unfavorable results. Examples of this are some tragic accidents that occurred in Tesla vehicles [31,32,33], highlighting the serious life-or-death consequences associated with the failures or miscalculations that can occur with vehicular systems. A Model X issued several audible and visual alerts before crashing into a concrete wall without the driver putting their hands on the wheel [31]. A preliminary report from the National Transportation Safety Council (NTSC) found that the driver of the vehicle activated autopilot, which provides automatic driving functions, 10 s before impact [32]. The document also indicates that the automatic system did not detect the driver’s hands on the wheel in the last eight seconds and that it did not perform evasive maneuvers to avoid colliding with a truck. In reference [33], the Tesla driver acknowledged that he was watching a movie on their mobile phone at the time of the accident. However, the vehicle never got to take control of the situation, necessary in view of the evident state of distraction of the driver. In this particular case, perhaps the first and foremost thing is to slow down the rush to “remove the human being” from the equation. Officially, Tesla’s autopilot was meant to aid the driver, not replace them. Human beings remain essential to driving and should continue to be so for some time until the technology matures. It is crucial to understand the surrounding cars with respect to the road context and to interact with them harmoniously for the success of the autonomous cars used in mixed urban traffic [34].

Similar examples can be evidenced by excluding any of the other two elements (vehicle and environment) of the system in the design of ADAS. The increase in road safety has motivated progress in the area of safety systems. Depending on the degree of automation of these, their action can be limited to driver assistance tasks or to be even of a more autonomous nature: acting directly on the components that modify the response of the vehicle, such as the acceleration and braking pedals and/or the power supply and steering control. The complex interactions between these characteristics and the actions of drivers have led to numerous investigations on the human factors involved in motor vehicle accidents. Some of these factors are demographics, distraction, experience, fatigue, alcohol, stress, the tendency to risk behavior, and decision making. At present, ADAS are a promising field of research in order to improve road safety.

While it is true that current projections on the levels of autonomy of modern vehicles tend towards a fully autonomous design, the fifth level according to SAE [35], the process of the mass inclusion on the roads of fully autonomous vehicles and their coexistence with human-driven vehicles is not expected to end in the short term [36]. Moreover, it should be noted that the very process of the development and improvement of the artificial intelligence systems that control autonomous vehicles requires for the continuous learning of the interactions between other drivers and road users, which is, in essence, learning to deal with human behavior. In the evolution of the levels of autonomy of vehicles, it is necessary to guarantee road safety in order to preserve lives and limit economic losses through accident avoidance. To this end, the implementation and development of an effective ADAS that accompanies and assists the evolution and maturity of autonomous driving is essential.

## 3. General Description of the Ideal Architecture of a C-ADAS

In this section, we describe our C-ADAS architecture proposal from a holistic and systemic vision. The systemic vision considers the relevance of the three main elements that make up the RSS (driver, environment, and vehicle). For its part, the holistic vision also reflects the relevance of considering the dependence and interrelationship that exists between each of these elements as essential requirements for the functioning of the system as a whole. Additionally, we establish a comparative analysis between this proposal and the architectures proposed in the previous reviews, describing its main characteristics and limitations. Figure 2 presents our C-ADAS architecture proposal.

This architecture considers the presence of the three main elements of an RSS as independent and closely interrelated subsystems, which influence and determine the state of road safety. The driver subsystem includes everything related to the actions and reactions of the drivers during the driving process, conditioned by their physical and psychological characteristics, such as age, gender, mental state, driving skills, experience, and driving profile. The vehicle subsystem considers aspects related to the electromechanical characteristics of the vehicle, braking behavior, acceleration, and sensing and communication technologies embedded on board, among others. Finally, the environment subsystem takes into account the rest of the elements outside the vehicle and the driver; among them, we can mention the elements of the road structure, the physical conditions of the road, the weather conditions, as well as other vehicles, pedestrians, animals, or obstacles on the road. We propose an integral communication system that seeks to leverage the advantages offered by the IoV concept, using data integration and the capacity for centralized decision making. Note that taking C-V2X communications into consideration implies a high degree of redundancy so as not to lose communication and to have several sources through which redundant data may arrive. Furthermore, the heterogeneity of networks and technologies may generate significant variations in the perceived delay and jitter among different communication paths, which become very sensitive issues for road safety applications. That is why a C-ADAS must conceive minimum delay times for the security modules, which could imply that in some cases the information that causes an action could be only conditioned by messages received from neighboring vehicles in the vicinity of the ego vehicle. In [37], an in-depth discussion of the control aspects of CAV systems is carried out, highlighting the main challenges generated by the existence of different information flow topologies, mainly focused on string stability, communication issues, and dynamics heterogeneity. These aspects must be taken into consideration for a practical implementation of a C-ADAS, such as the one proposed here, since the heterogeneity in the network topology and the high dynamism of vehicular networks can impose additional communication challenges. However, we are committed to the idea of conceptualizing the use of on-board communication devices with C-V2X functionality, framed within the IoV concept, given that current and future developments point to the convergence of technologies. In this sense, 5G is seen as a key technology to support the development of future vehicular networks, and presents low-latency values in communications, which will undoubtedly help to compensate for the delays associated with the processing and control of communication schemes.

This architecture includes three modules to capture the data from the bidirectional interaction between the three main elements of the RSS, these are the natural perception module, the assisted perception module, and the human–machine interface module. Additionally, a main module is included in an upper layer that receives, processes, and stores the collected information by the lower modules to make decisions about the state of road safety. This is a modular architecture where the main module can be implemented both locally, inside the vehicle, and remotely, in a centralized management station within the environment of intelligent transportation systems (ITS) under the IoV concept. In the latter case, the entire flow of information from the three data capture modules would be shared through the V2X communication devices on board the vehicle. From the review of the state of the art in articles of enabling technologies, we can divide them into sensor technologies, wearable technologies, and communication technologies, which are related to each of the data capture modules and their connections with the main module. Although the most researched enabling technologies are those that intervene in the vehicle–environment interaction, it is also important to highlight those that are part of the vehicle–driver and driver–environment interactions, since all of them together allow for a closing of the cycle in a holistic and systematic design of cooperative ADAS, where the three main elements of an RSS are present: driver, vehicle, and environment. In this sense, we have tried to list below the most relevant technologies (described repeatedly in many of the articles consulted), but at the same time to reflect, through an integrating spirit, the technologies present in each of the three areas of interaction between the three main elements of the RSS. For a greater level of detail, the reader can consult these technologies in references [23,25,27,29,38,39].

(i) Sensor technologies:Radar systems are classified into: (i) short-range radars (SRR), which have a detection range of up to 20 m, are based on a single antenna, and are not capable of detecting angular information; (ii) long-range radars (LRR), which have a range of up to 150 m and an angular resolution of up to 2 degrees [40];Cameras embedded on vehicles are of two fundamental types: (i) stereo cameras used to obtain a wide panoramic vision in conditions of good visibility; (ii) infrared cameras used in situations of reduced visibility at night or in the presence of adverse weather conditions [41];Light detection and ranging (LIDAR) uses laser signals to determine the relative distance of nearby objects from a vehicle. Laser signals are emitted and their respective echo signals are received to calculate these distances, with detection ranges between 10 and 200 m [42];Acoustic sensors use an operating principle similar to that of radars and LIDAR, but they use high-frequency sound waves (ultrasonic) to determine the distance of an object to the vehicle [43].

(ii) Wearable technologies (electronic devices designed to attach to the user’s body—classification depends largely on the device’s functional properties):Smartwatches are electronic devices with functionalities such as GPS, fitness/health monitoring, and waterproof operation [44];Wearable cameras are much more flexible and mobile than conventional cameras, as they focus on a first-person view and are often attached to eyeglasses, helmets, and caps [45];Smart eyewear are used to provide information, notifications, and a three-dimensional view through optical head-mounted displays (OHMDs), heads-up displays (HUDs), virtual reality (VR), augmented reality (AR), and/or mixed reality (MR) [45];Fitness trackers are placed on different parts of the body to monitor the physical state of the individual during the performance of daily activities or exercise routines. Their measurements include parameters such as speed, heart rate, calories released, and number of steps [46];Smart clothing—usually shoes, hats, clothing, and helmets—incorporate cameras and sensors to monitor body signals and adapt their characteristics to the individual’s state [47];Wearable medical devices made up of one or several biosensors are used to monitor the physiological activity of the individual for the purposes of the prevention, diagnosis, and early treatment of diseases and health status abnormalities by measuring temperature, heart rate, blood pressure, and glucose level, or by performing electrocardiography (ECG), electroencephalography (EEG), or electromyography (EMG) [48].

(iii) Communication technologies:Dedicated short-range communication (DSRC) is a technology of fully integrated vehicular networking, implemented over the 75 MHz bandwidth (5.85–5.925 GHz) assigned for the Federal Communications Commission (FCC). The architecture and the services to enable these secure V2V and V2I communications in the Wireless Access Vehicular Environment (WAVE) are provided by the IEEE 802.11p and IEEE 1609 protocol suites [49];Light fidelity (Li-Fi) uses wireless communications in the visible-light band for data transmission by encoding the flashing states of light-emission diodes (LEDs) [50];LTE Advanced Pro (LTE-A) is part of the evolution of LTE networks in version 14 to ensure that V2X service requirements are supported by the LTE transport network. Different V2X application scenarios are defined, including V2V, V2I, vehicle to pedestrian (V2P), and vehicle to network (V2N) [51];The “5G/IMT-2020 Standing Committee” reports the standardization process of fifth-generation wireless communication technologies (5G) [52]. This technology is expected to be the future of vehicular communication networks, providing support for low-latency and ultra-reliable communications (URLLC) scenarios.The internal communication networks of a vehicle are composed of electronic control units (ECUs), mechanical and electric sensors, and actuation devices to guarantee the correct vehicle operation. Original equipment manufacturers (OEMs) today design proprietary devices and networks to share data through on-board diagnostic (OBD) hardware. Well-established technologies such as CAN bus, MOST, LIN, and FlexRay are examples of their resilience and flexibility. Other emerging technologies such as the vehicular Ethernet support the growing communication capacity demanded for modern vehicles, and are discussed in reference [53].

The main module is in charge of using the information from the three data capture modules to analyze the state of road safety and make the best decision when it comes to assisting the driver or acting directly on the vehicle’s safety systems in case of the non-attention of the driver regarding warning notices to an imminent danger situation, with the aim to minimize the level of risk and traffic accidents. It is also in charge of sharing this information with the other actors in the road environment. The operation of this module begins with a reception information block that receives the data from the three capture modules. Once this data has been processed, models of driver behavior are established and stored, and customized according to the characteristics of the driver. These models allow for estimating the prediction of the longitudinal and lateral movement of the vehicle, information that can be used by different advanced driver assistance systems (ADAS-1 … ADAS-N blocks) either to provide assistance to the driver or to take direct control of the vehicle in case the driver does not respond adequately and in a timely manner to certain dangerous situations on the road. In this sense, the decision making of the C-ADAS system may entail carrying out the functions of vehicle control, visualization through the HMI, or the transmission of information related to road safety to the rest of the actors on the road or to the centralized management station in an ITS.

Under the IoV concept, different multi-layer architectures have been proposed and different types of interactions between the elements that compose it have been established [17,54]. One of the layers that is commonly addressed is the centralized platform in the cloud (associated with the “intelligent brain” in the IoV architecture). In this, the centralized processing of the road safety data obtained in the lower layers is carried out, as well as the global strategies and the management of road safety alerts, assisted automatic driving, and intelligent navigation, among other functions. Centralized or distributed execution implies differences in the signaling overhead and delay associated with decision making, but the processing delay in the cloud is less than in the computer equipment inside the vehicle. Some critical security applications will demand URLLC-type communications, which will benefit from the growing development of 5G. The analysis of the local data allows to guarantee the control of the main functions of the vehicle in real time, while the remote analysis of the same allows to create added value in the system to improve the reliability, the efficiency, and the performance of the vehicle through the application of communications, computational processing, and distributed information on a large scale. To the extent that connectivity capabilities and computer technologies are developed and latency in the execution of applications is reduced, the use of aggregated data, obtained remotely, becomes more useful in the operation of these road safety applications in real time. Vehicles do have limited computation and storage resources that may not be sufficient for road safety applications that need to process a big amount of data. Since they require big storage and complex computations, for this, vehicle-to-cloud computation helps by providing proficient support to these applications. In our architecture, the main module is flexible in order to operate with some of these functions, and is mainly oriented to the design of advanced driver assistance systems. Another layer that is described in an IoV architecture is the data acquisition layer, whose main function is to collect different types of data from different sources and to digitize the data to ensure that it can be successfully transmitted and analyzed. In our architecture, this IoV layer is represented by the three data capture modules: the natural perception module, the assisted perception module, and the HMI module, described below.

The natural perception module is responsible for capturing the elements of the D–E interaction during the natural perception process from the driver of the surrounding environment. Through devices implemented inside the vehicle, such as vision cameras, heart-rate-monitoring devices, and ethyl breath detection, the driver’s behavior can be analyzed based on the monitoring of their activity on board and their reactions to the dynamics of the environment, which can help infer their behavior and future actions on the road [21].

The assisted perception module is responsible for capturing the elements of V–E interaction through sensors, such as as radars, ultrasonic, GPS, IMU, LIDAR, and cameras, jointly with communications equipment, meaning that the state of the surrounding environment of the vehicle is sensed and its dynamic data are captured for the posterior presentation to the driver as additional information of their natural perception. This module is also responsible for communicating the driver’s mobility and status information to the environment while driving, as well as ADAS alerts and warnings and the driver’s reaction to them.

The HMI module is responsible for capturing the elements of the D–V interactions during the driving process. It is composed of the human–machine (H-M) visualization devices, such as displays, vision cameras, or instrument panels, and by the H–M control devices, such as the acceleration, brake, and clutch pedals, the steering wheel, switches, panel buttons, and the gear lever, etc. The four principal functions of this module are: (i) sensing the physical actions of the drivers over the direct and indirect devices involved in the driving process, such as the steering wheel, the brake, clutch, and acceleration pedals, the transmission state, lights, and other on-board controls; (ii) displaying information about the vehicle status (from the internal modules of the CAN bus and GPS modules) and the surrounding environment (maps, routes, traffic signs, location of other vehicles, pedestrians, and obstacles on the road); (iii) visualizing alerts for the driver regarding dangerous situations as a product of the analysis of data related to road safety; and (iv) sensing the driver’s reactions to the alerts emitted by the assistance system.

Control systems refer to where C-ADAS processing and decision-making takes place, it can be locally in the vehicle itself or remotely, under IoV concept. The modular structure describes the elements that make up the system and their functions. Personalization refers to the degree of mutual adaptation between the system and the driver during the vehicle-driver interaction, it includes the latter’s actions during the driving process, as well as its reaction to the system alert notices presented through the HMI interface of the vehicle. Cooperative communication highlights the capacity of the system to communicate to other actors in the road environment about the information regarding the driver’s actions and the result of the system’s decision-making regarding the state of road safety. The assistance functions basically refer to assisting the driver or taking control of the vehicle’s safety systems. It also highlights the system’s ability to operate simultaneously with more than one ADAS (LCA, FCW, AEB, BSW, LDA, among others).

However, monitoring the progress of these technologies, as well as the achievements and projections of the main automakers, will favor the implementation of a proposal close to this C-ADAS that efficiently exploits bidirectional interaction between the three subsystems to increase road safety.

Table 2 resumes the analysis of the architectures proposed in the surveys consulted and our architecture proposal, highlighting the control systems, the assistance functions, and the degree of personalization and cooperation, as well as the main elements that compose them, describing also how the interactions between these elements are implemented.

## 4. Vehicle-Environment (V–E) Interaction

The analysis of bidirectionality in the study of the interaction between driver and environment (V–E) is relevant to all ADAS. If we analyze, for example, the information coming from the environment, we find that it can be received by the C-ADAS through several ways: (i) directly, through the information captured by the set of sensors and communication devices that are found on board the vehicle (E -> V) or else (ii) indirectly, at first through the information captured by the driver through their sensory elements and later, through the information partially captured by devices on board the vehicle in charge of monitoring the status and behavior of the driver as a reaction to the stimuli that he perceives from the environment (E -> D -> V). This alternative path of redundancy is relevant to consider, given that sensors and communication devices present limitations and challenges for their optimal operation. This degree of redundancy can only be achieved if the design of the C-ADAS is approached from a holistic and systemic perspective, where the study of directionality in interactions is considered. The works of the state of the art that are described in this section, fundamentally contribute to the first of these two ways (directly).

The main elements associated with the assisted perception module of the surrounding environment are described in this section, as well as the signaling devices, sensors, and communication technologies involved with the operation of this module. On the one hand, this perception “assists” the driver in acquiring information from the environment in which he/she operates. On the other hand, the vehicle, through its signaling and communication devices, communicates to the surrounding environment information on its movement state and also information related to the intention and actions of the driver. In a general sense, the sensors and communication technologies improve the range and precision of the information that the driver can perceive, especially in variables that are difficult to estimate, such as distances, speeds, and the relative accelerations between vehicles and other actors in the environment.

With the use of communications, high-level information can be incorporated with a high predictive degree, which allows for the anticipation of changes in the dynamics of vehicle movement before they materialize and can be perceived by drivers. In the same way, the range of perception coverage can be extended and information can be obtained beyond the local environment of the vehicle, which is associated with scenarios where there are no direct-line-of-sight conditions.

### 4.1. Actual State

The discussion of this section begins by grouping the works that use vision and infrared cameras to capture the information from the vehicle–environment interaction, a method that is currently widely used in the automotive industry, but with limitations related to adverse weather conditions, lighting deficiencies, and the high computational cost of image processing algorithms, to name a few. Xin et al. [55] propose an intention-conscious model to predict the trajectory based on the estimated lane change intention of neighboring vehicles, using an architecture with two long–short-term memory (LSTM) networks. The first one receives as input sequential data that characterizes the lateral movement of the vehicle to infer the driver’s intention to stay in the lane, turn left, or turn right. Once the target lane is detected, this indicator is passed as an input to the second LSTM network, which also receives the sequential data of the longitudinal movement of the vehicle, to finally predict its position. From the view point of the ego vehicle, only the features that can be feasibly measured using on-board sensors, such as LIDAR and radar, are used as input. The database used in this paper is from the next generation simulation (NGSIM) [56].

Deo et al. [57] design an LSTM encoder–decoder model that uses convolutional social-grouping layers as an enhancement of social-grouping layers for the robust learning of inter-dependencies in vehicle movement. The social grouping is defined by a structure called the social tensor, which groups the LSTM states of all the agents located around the predicted agent. This is done by defining a spatial grid around the agent being predicted and filling the grid with LSTM states based on the spatial configuration of the agents in the scene. The encoder is an LSTM network with shared weights that learns vehicle dynamics based on trajectory histories. The output of the LSTM decoder generates a probability distribution on future movement for six maneuver classes and assigns a probability to each maneuver class. A lot of complementary information can be captured using visual- and map-based cues. For the experiments, they use the publicly available NGSIM database. Deo et al. [58] propose a variation of the architecture designed in [57], using an LSTM network as an intermediate layer to classify and assign a probability to the maneuvers instead of the convolutional social-grouping layer. The results of these analyses evidenced the importance of modeling the movement of adjacent vehicles to predict the future movement of a given vehicle, as well as the importance of detecting and exploiting common vehicle maneuvers for the prediction of future movement. They use the publicly available NGSIM database for the experiments.

Kim et al. [59] present a collision risk assessment algorithm that quantitatively assesses collision risks for a set of local trajectories through the lane-based probabilistic motion prediction of surrounding vehicles. Initially, the target lane probabilities are calculated, representing the probability that a driver will drive or move into each lane, based on lateral position and lateral velocity in curvilinear coordinates. It assumes that the lateral offset of vehicles with respect to the road center-line is measured from a suitable sensor suite, such as a camera, radar, or LIDAR. To estimate the collision probability, the collision risk is assumed as a metric, which is modeled as an exponential distribution, dependent on the time to collision (TTC). The prediction performance of the lane-based probabilistic model is first validated by comparing the model probabilities from the probabilistic target lane detection algorithm against the maneuver probabilities obtained from real-world traffic data from the NGSIM database.

Hou et al. [60] propose a model of mandatory lane change (maneuver of incorporation into the vehicular flow of a highway) that considers as input variables data associated with the distances and relative speeds between the vehicle that is going to carry out the maneuver and the front and rear vehicles in the lane to which it is intended to enter, in addition to the distance the vehicle has traveled on the merging lane. Bayesian classifiers and decision trees are used to predict the driver’s decision to carry out the maneuver or not, determining as the most relevant variable the relative speed between the vehicle carrying out the maneuver and the vehicle in front on the line to which it is intended to enter and, in general, the greater relevance of relative speeds over relative distances. Detailed vehicle trajectory data from the NGSIM database were used for model development (data of U.S. Highway 101) and testing (data of Inter-state 80). Liu et al. [61] develop a deep learning model to evaluate discretionary lane change maneuver decision making. The model is based on deep neural networks and with the exception of the instantaneous states of the subject and the surrounding vehicles, the historical experience of the drivers and the memory effect from vehicle to vehicle are also taken into account for the final evaluation of the maneuvering situation of change of lane, considering the analysis of the time series of trajectory data as part of the historical behavior of drivers. The classifier used is a gated recurrent unit (GRU) neural network, which is a type of RNN. They use the traffic data of the NGSIM database to train and test the model.

Benterki et al. [62] present a system for predicting lane change maneuvers on motorways. These maneuvers are classified into left turn, right turn, and lane keeping, using two machine learning techniques: a support vector machine (SVM) and neural networks. The system also estimates, with a time window in advance, the time in which the lane change maneuvers will take place. The lane change process is subdivided into three stages: preparation of the lane change, active execution of the lane change, and completion of the lane change. Therefore, the system proposes to exploit the changes that occurred during the lane change preparation stage for the premature detection of maneuvers. The real-driving data of the NGSIM database is used for training and testing. Ding et al. [63] propose a method that combines high-level policy anticipation with low-level context reasoning. An LSTM network is used to anticipate the vehicle’s driving policy (go ahead, yield, turn left, and turn right) using its sequential historical observations. This policy is used to guide a low-level optimization-based reasoning process. In this reasoning process, cost maps are defined to represent the context information, which are associated with certain characteristics of the road, such as lane geometry, static objects, moving objects, the area enabled for driving, and speed limits. The open-source urban autonomous driving simulator, CAR Learning to Act (CARLA) [64] is adopted to collect the driving data, with the use of a Logitech G29 racing wheel.

Mahjoub et al. [65] propose a stochastic hybrid system with a cumulative relevant history based on GPs. This design is used within the context of model-based communication to jointly model driver/vehicle behavior as a stochastic object and obtain accurate predictive models for mixed driver/vehicle behavior trends in the short and long term (within 0–3 s) of the critical dynamic states of the vehicle, such as its position, speed, and acceleration, within the discrete modes of the system, which are equivalent to the different long-term behaviors (maneuvers) of the driver. The lane change maneuver is selected as a specific long-term driver behavior, and the lateral position of the vehicle is modeled through an available set of already-observed instances. This is done by building a cumulative training history of on-the-go maneuver-specific data from identical or relevant maneuvers observed in the driver’s recent driving history, and then feeding this training data to the model inference block, such as your initial training set. To evaluate the proposed method, real trajectory data of 40 lane change maneuvers from the NGSIM database were used. As a recommendation for network situations with a high degree of congestion, where frequent reception of messages is difficult, a model that combines the constant speed model with the proposed Gaussian regression model would ensure prediction for both the near future (less than one second), as well as for the far future (between one and three seconds).

The following describes the works that use more specific sensors of greater complexity and economic cost, such as radar, LIDAR, GPS, and IMU, to capture the information from the interaction with the vehicle environment. These devices, on the other hand, have disadvantages due to direct-line-of-sight obstruction problems, in the case of LIDAR radars, and in general to unfavorable environmental conditions. Batsch et al. [66] propose a classification model using a Gaussian process (GP) for the problem of detecting the presence or absence of the risk of collision between a vehicle and the vehicle that precedes it, which circulates at a slower speed as a result of being part of a traffic congestion scenario that includes several vehicles. To train the system they use data produced by CarMaker simulation software [67]. The tests are conducted in an automated vehicle equipped with a radar sensor, neglecting the sensor uncertainty in the velocity and aperture angle measurement. Zyner et al. [68] present a method based on recurrent neural networks to predict driver intention by predicting multi-modal trajectories that consider a level of uncertainty. To deal with data sequences of different lengths, sequence-fill techniques are introduced, taking as reference the last known position of the vehicle. The data analyzed contains the lateral and longitudinal position track history, as well as heading and velocity. Park et al. [69] employ the use of LSTM networks to predict the future trajectory of surrounding vehicles based on a history of their past trajectory, formulating the vehicle trajectory prediction task as a multi-class sequential classification problem. For the evaluation of the system, real-vehicle trajectory data from a highway environment was employed. To capture the vehicle–environment interaction data, the test vehicle used a radar sensors and the IMU sensors.

Next, the works that use, together with the use of cameras, the incorporation of sensors, such as radar, LIDAR, GPS, and IMU are grouped to capture the information from the interactions with the vehicle environment. It is worth noting the fact that by using a greater number of sensors of different technologies, a greater degree of robustness of the system is achieved due to the redundancy in the information that can be received, but on the other hand, a greater degree of processing is necessary for data from diverse heterogeneous sources, which increases the computational cost. Liu et al. [70] establish an autonomous lane change (discretionary maneuvering) decision-making model based on benefit, safety, and tolerance functions that analyze not only lane change factors in autonomous vehicles associated with route planning and monitoring, but also in addition to the lane change decision-making process. The benefit function considers the relative speed and distance data between the vehicle and the predecessor vehicles in the same lane and the target lane. The safety function considers a minimum safe distance between the vehicle and the successor vehicle located in the target lane, in addition to the relative distance and speed values between the two. Finally, the tolerance function considers relative distance and speed values between the vehicle and the predecessor vehicle in the same lane, avoiding frequent lane changes if the distance between them is too great. In order to verify the effectiveness of the model in real scenarios, they realized a test verification in the vehicle. The test vehicle used is equipped with Mobileye, millimeter-wave radar, mobile station GPS, AutoBox dSPACE, IMU, and other devices.

In this site are grouped the works that exclusively employ the use of vehicular communications (DSCR) to capture the information from the interactions with the vehicle environment. Although the use of communications manages to fundamentally compensate for the range limitations presented by the use of sensors and cameras, providing greater flexibility in terms of the road safety information that can be exchanged, there are also a series of limitations associated with all technologies of wireless communication, such as packet loss, transmission errors, and communication delay, to name a few. Fallah et al. [71] use the model-based communication scheme in a cooperative FCW system using the example of CAMPLinear and, specifically, the collision detection algorithm proposed in [72] and later refined in [73]. This algorithm uses speed and acceleration information as input, both from the host vehicle and from the remote vehicle, which is the vehicle ahead in its own lane. The model used to estimate the remote vehicle mobility data belongs to the family of follower car models; specifically, the model introduced in [74] is used. The concept of hybrid automation is used [75], which is a well-known method for modeling mixed systems of discrete and continuous states. To evaluate the model-based communications (MBC) approach, two configurations are presented (MBC1 and MBC2), which are compared with traditional communication schemes that directly transmit speed and acceleration data. In both MBC1 and MBC2, each vehicle sends its movement models once at the start of the test and then periodically transmits the updates of the model inputs (speed and acceleration). In the case of MBC2, more sporadic additional messages associated with the change in the movement pattern are also transmitted. It is precisely this second configuration that obtains the most accurate results when tracking the movement of a vehicle.

Huang et al. propose [76] and develop [77] a mechanism based on the real-time estimation of the position-tracking errors of neighboring vehicles, to manage the cooperative information exchange in the vehicle communications environment. They first evaluate decentralized information dissemination policies for tracking-error-dependent multiple dynamic systems and then use collision-error-dependent policy to obtain better tracking performance. Finally, the transmission probability is calculated for each vehicle every 50 milliseconds based on expected tracking errors. Upon receiving information from the channel, each vehicle updates its estimated states of the neighboring vehicles using a first-order kinematic model, i.e., a constant speed predictor. The main concept of measurements in these algorithms is that the generation of messages from the receiver and the timing of the communication must be determined so that the position-tracking error is reasonably limited. A local copy of the neighboring estimators is executed at each local estimator. The sender compares the output of this simulated estimator with its actual state, thus estimating the position-tracking error, and determines whether remote vehicles need updated messages from the sender. The decision is made by comparing the position-tracking error with a configurable error threshold, generally defined according to the requirements of the cooperative road safety application (RSA).

Mahjoub et al. [78] propose a technology-independent hybrid model selection policy, based on the MBC scheme, for vehicle-to-everything (V2X) communication. The core idea is to implement a hybrid modeling architecture that switches between different modeling subsystems to adapt to the dynamic state of the vehicle. In this particular case, two modeling states are used: one governed by GP, which uses two kernels: one linear and one radial basis function. The other modeling state is defined by the constant velocity kinematic model. A tracking error threshold is used as a selection element when using these models. The results show the effectiveness of the proposed communication architecture both in reducing the required message exchange rate and in increasing the accuracy of remote vehicle tracking. The greater tracking accuracy of the MBC scheme can be attributed to its ability to capture higher-order vehicle dynamics as a result of harsh braking maneuvers and lane change maneuvers.

Mahjoub et al. [79] explore the modeling capabilities of the non-parametric Bayesian inference method: GP, integrated into the MBC design scheme, accurately represents different patterns of driving behavior using only a bank of GP kernels of limited size. To do this, a group of representative trajectories from the SPMD data [80] were selected and the properties of the required kernel bank were explored to be modeled within the GP-MBC scheme. The two fundamental metrics used to evaluate the proposed system are: the length of the transmitted message (related to the size of the kernel bank) and the message transmission rate (related to the persistence of the model). The existence of such a kernel bank allows for transmitting entities to send only the kernel ID instead of the kernel itself, which consequently reduces the length of the packet. The persistence of the model is understood as the time in which a model remains valid for the prediction, i.e., obtaining a margin of error in the prediction lower than the threshold established by the RSA requirements. The results obtained showed the feasibility of using a group of GP kernels of finite size to predict, with the precision required by the RSA, the future position of the vehicle through an indirect prediction method, i.e., by predicting the values of the time series of the future speed and direction of the vehicle. An indirect position estimation (speed, direction, or acceleration) achieves superior results compared to direct position estimation.

Vinel et al. [81] design an analytical framework that considers the behaviors of cooperative road safety applications considering the performance of V2V communications. The relationship between the characteristics of V2V communications associated with the probability of packet loss and the packet transmission delay, with the physical mobility characteristics of the vehicle, such as the inter-vehicle safety distance, is analyzed. The case of the cooperative ASV of the emergency electronic brake light defined by the ETSI is analyzed.

Finally, the jobs that use, together with the use of vehicular communications (DSCR), cameras and sensors such as radar, LIDAR, GPS, and IMU are grouped to capture the information from the interaction with the vehicle environment. This favors the complement between communication and sensor technologies, taking the best of both and guaranteeing a more complete performance and greater possibilities for facing the challenges in the area of road safety. Mahjoub et al. [82] design a system for the prediction of the lateral and longitudinal movement of the vehicle. For the prediction of the longitudinal trajectory, nonlinear auto-regressive exogenous models based on neural networks are used. In the case of lateral trajectory prediction, recurrent neural networks (RNN) are used. The system uses two main sources of information: (i) cameras and on-board detection devices such as radars and LIDAR that are assumed as the primary information providers for CAV applications; (ii) V2V communication, which is obtainable using dedicated short-range communication (DSRC) devices, and is regarded as an important supplementary information source whenever it is accessible. The performance of the system is evaluated not only in ideal communication conditions, but also in the presence of scenarios with up to 40% losses. In this case, a zero-hold estimation method is included to combat packet loss or sensor failure and to reconstruct the time series of vehicle parameters at a predetermined frequency. Its evaluation is simulated using real-communication scenarios with data extracted from safety pilot model deployment (SPMD) [80]. Du et al. [83] develop a network architecture called vehicular fog computing to implement the cooperative data census of multiple adjacent vehicles circulating in the form of a platoon. Based on this architecture, a greedy algorithm is used to maximize the census coverage (associated with the area ratio and total ratio parameters) and minimize the overlap coverage (associated with the efficiency parameter), enhancing the parallel calculation through of the distributed management of the computational resources of the platoon members. A SVM algorithm is used to merge the census data of multiple vehicles and obtain precise information on the status of the vehicles. Through an occupancy grid filtering (OGF) of the on-board sensors (LIDAR, cameras), the environment is mapped as occupancy states. These OGF maps are integrated into the head vehicle, and by means of SVM it is classified when a grid is occupied by a vehicle. To train the SVM classifier, the GPS position data extracted from the NGSIM database is used to locate the vehicles on the OGF map. Finally, the result of the merger of the census data of multiple vehicles (location of the vehicles on the OGF map) feeds the algorithm of a light GRU neural network to predict the discretionary maneuvers of lane changes to classify them into lane-keeping maneuvers or lane change maneuvers.

Moradi-Pari et al. [84] use the model-based communications scheme to design a small-scale and large-scale modeling strategy for the dynamics of vehicular movement. The representation model of the system to describe the behavior of the vehicle is based on the representation of stochastic hybrid systems, where: (i) small-scale evolution represents actions of braking and acceleration, represented by exogenous auto-regressive models and (ii) large-scale evolution, which includes lane change maneuvers and free circulation flows, are represented by coupling these models within a Markov chain. At each model calculation time, all currently available states of the latest version of the model must be explored, and the best-fit parameter values must be found for each of them according to the new observation element. If at least one of these states satisfies the error threshold specified by the application using its new parameter values, the current model is assumed to be fully descriptive for the entire observation sequence received. However, if the minimum error reached given by the current model exceeds the required threshold, it is necessary to introduce a new state to represent the new observation segment and describe the last maneuver of the driver. To evaluate the performance of the proposed models and adaptive cruise control methods, various scenarios were simulated with different realistic data sets, including data from SPMD [80] and driving cycles for Environmental Protection Agency testing standards [85].

We want to emphasize that cooperative driving can significantly contribute to the development of C-ADAS, since this is the most important result of detection, communication, and automation technologies, and, in turn, significantly influences the behavior of drivers. Z. Wang et al. [37] provide a review of the literature associated with cooperative multiple CAV longitudinal motion control systems, with an emphasis on the architecture of several cooperative CAV systems. An in-depth discussion of the control aspects of CAV systems is carried out, highlighting the main challenges generated by the existence of different information flow topologies, which are mainly focused on string stability, communication issues, and dynamics heterogeneity. Zhou et al. [86] present a literature review of learning-based longitudinal motion planning models for autonomous vehicles, focused on the impact of these models on traffic congestion. They surveyed the non-imitation learning method and imitation learning method, and the emerging technologies used by the principal automakers for implementing cooperative driving are described.

Multiple research works have investigated in the context of cooperative driving the design of control systems that favor the management of traffic and/or the crossing of intersections. Zu et al. [87] propose a cooperative method for connected automated vehicles that controls the timing of the traffic lights and manages the optimal speed at which the vehicles should circulate. The optimization of the traffic light times and the calculation of the arrival times of the vehicles at the intersection allows for the minimization of the total travel time for all the vehicles, as well as the fuel consumption of the individual vehicles. Zheng et al. [88] establish analytical results on the degree of stability, controllability, and accessibility of a mixed-traffic system composed of autonomous vehicles and human-driven vehicles. The proposed system allows the flow of traffic to circulate at a higher speed and shows that the autonomous vehicles, along with cooperative driving, can save time and energy, smoothing traffic flow and reducing traffic undulations. Wang et al. [89] propose a cooperative platoon system for CAV, based on a predictive control model with real-time operation capability, to efficiently manage the vehicle tracking behaviors of all CAVs in a platoon. The constant time advance method is used to adjust the balance gap between successive vehicles. Zhou et al. [90] introduce a smooth-switching control-based cooperative adaptive cruise control scheme with information flow topology optimization to improve riding comfort while maintaining string stability. A Kalmann filter-based predictor is used to estimate the state of the preceding vehicle, suppressing the noise in the measurement and estimating the acceleration of the vehicle in the event of communication failures. Zhou et al. [91] propose a hybrid cooperative intersection control framework to manage the entrance and exit of a group of vehicles to an intersection. A virtual platoon is defined to group these vehicles according to their proximity to the entrance of the conflict zone of the intersection. The location assignment of the vehicles within the virtual platoon differs from their real relative locations. This virtual platoon is obtained by linearly projecting the distances at which the vehicles are from the center of the intersection, then, platoon control rules are applied to manage the movement of vehicles approaching the intersection.

The management of the flow of information exchanged in a V2V communication environment has been addressed in [92]. Wang et al. develop a mathematical modeling based on queues to manage the transmission of information of multiple classes with different levels of delay according to the cooperative road safety applications. In [93], these authors also addressed traffic management in mixed environments with the presence of human-driven vehicles and autonomous and connected vehicles, analyzing the differences in the principles of route choice and the traffic patterns followed by human-driven vehicles and connected and autonomous vehicles, and they also considered the use of preferential circulation lanes with free access to connected and autonomous vehicles in [94].

Table 3 summarizes the main works consulted that address elements of the vehicle–environment interaction. The directionality in which this interaction is approached is analyzed, as well as the way in which it is implemented.

### 4.2. Current Challenges

The addressed challenges in this section are focused on the sensors and communications technologies. The main challenges with regard to sensors technologies are:

(i) The sensor occlusion with respect to the line of sight of the objects and other road actors. The good performance of these technologies is affected under various climatic and environmental conditions, such as roads with markings covered by snow, heavy rain, or dense fog. Objects, people, and animals located in the vicinity of the vehicle, or obstructing each other, represent serious security problems for detection by these devices. This phenomenon is not so serious when the objects are located at greater distances, where the processing algorithms can help the sensing devices to improve detection tasks. These drawbacks can be minimized by sensor redundancy as in [95], where a 360-degree vision system is used for parking assistance.

(ii) The high computational resource consumption of image processing algorithms present in camera-based sensors. Detection at distances greater than 200 meters requires the use of ultra-high-resolution cameras for the sensing of small details in the target image. Therefore, powerful image processing algorithms are required to analyze the high volume of image data and extract useful information from the noise associated with it [96], which continues to represent a limitation to the adequate processing in real time that road safety demands.

(iii) The high cost of specific hardware technologies. Some well-established technologies in the market, such as vision cameras or radars, have managed to establish themselves in large-scale production, lowering their production costs. This situation, however, differs from other more specific technologies for the automotive industry, such as LIDAR devices, whose standard incorporation in vehicles considerably increases their price depending on various factors, such as the type of use for the one that the vehicle is destined for, or their sensing capabilities in 2D or 3D.

The main challenges with regard to communications technologies are:

(i) The scarce deployment of network infrastructure in the road environment. The massive implementation of V2I technologies is an expensive and time-consuming task. Various costs must be assumed depending on the location environment, such as the installation of nodes, bandwidth, and energy support, and the subsequent maintenance of the installed equipment. An important aspect is the traffic capacity of these networks, given that cooperative road safety applications require a high degree of penetration of vehicles with installed connection capabilities, e.g., in the case of the cooperative collision warning application, a density greater than 60% of the total connected vehicles is desirable [97].

(ii) The lack of robustness of current communications networks to operate in a vehicular environment, evidenced by the loss of transmitted packets and other problems, such as communication channel congestion, transmission delay, and fading and shadowing in signal propagation. The reliability and accuracy of the exchanged data is vital to ensure the proper functioning of cooperative applications, as inaccurate data can result in bad judgment when making a decision related to road safety. These networks must guarantee robustness against sending duplicate messages or false positives in the issuance of alerts. Furthermore, even when the data exchanged is accurate, the freshness of the information is required, due to the importance of the temporal component of road safety data. It is not enough to detect a risk situation on the road and communicate it accurately, it is also required that this information arrives in time for the appropriate reaction to said danger and thus guarantee the good global operation of cooperative road safety applications. In this sense, emerging technologies such as 5G can provide greater capacity and reduced latency when exchanging this sensitive information [98]. With the progressive increase in the volume of data, the need arises to have tools and algorithms capable of efficiently processing them. Raw data handling is becoming less feasible and more expensive. Dimensionality reduction techniques are required to identify patterns in data and achieve more scalable developments [99].

(iii) The security and privacy of the information exchanged represent an important challenge to consider when designing C-ADAS. As in any communication network where private information is exchanged, the issue of data security and integrity is vital, but in these networks in particular it is even more important, since people’s lives are at stake, as well as overall road safety. Hacking activity in networks without adequate protection can cause attackers to take control of a vehicle’s security systems, causing traffic diversions or the activation of the emergency braking system, the stealing of personal data from users, and in the worst case scenario, the creation of conditions for the occurrence of traffic accidents. These challenges are part of the OEM and after-sales connectivity systems [100], which is especially sensitive in automated vehicles with electronic control of their actuators. Another current problem is that network security approaches can, on occasion, compromise vehicle security due to the overload of exchanged security information, generating excessive delays in communications, which is associated with authentication mechanisms, validation, and security certificates.

The shortcomings that still persist, and the study and achievements related to the V–E interaction, are the inability to operate bidirectionally, the lack of holistic integration between the detection, communication, and processing technologies, as well as the joint operation with the driver subsystem, which allows for the exploitation of these redundant pathways to obtain information from the environment. In addition to this, there are also the challenges associated with the functioning and operation of these sensing (e.g., resolution and scope) and communication (e.g., reliability and delay) technologies, which undoubtedly limit the effectiveness of the current implementation of the proposed C-ADAS.

## 5. Driver–Vehicle (D–V) Interaction

The analysis of bidirectionality when studying the interactions between driver and vehicle (D–V) is relevant for proactive decision making in a vehicular context. Let us consider an example where the driver is developing aggressive driving on the road—the C-ADAS detects this behavior (D -> V interaction). This characterization of the driver will allow the C-ADAS to estimate possible risk situations when this vehicle approaches other aggressive or extremely conservative drivers. Given such knowledge, convenient and personalized messages could be sent to moderate the conduct of those involved, which is a more complex form of interaction (D -> E -> V). This redundancy alternative path is relevant to consider, given that HMI devices have limitations and challenges for optimal performance in cooperative environments and sometimes limited customization features. This degree of redundancy can only be achieved if the design of the C-ADAS is approached from a holistic and systemic vision, with bidirectional interactions among subsystems.

The D–V interaction describes the main elements related to the HMI module. This area is in charge of presenting the driver with the information about the environment acquired by the sensors and communication technologies and the information of the vehicle itself, obtained from the internal communication pathways, such as the CAN bus. Similarly, it is also necessary to sense the driver’s reaction to the operation of the assistance system. At the same time it is in charge of materializing, through the vehicle operation, the driver’s actions on the environment that surrounds him. As an “assistance” system, its first function should be to learn about the particular characteristics of the driver, understand their actions, and to be able to model their behavior (instantaneous and historical) to identify the “particular assistance needs” that the driver requires. It must also be able to communicate that knowledge acquired to the rest of the actors on the road, as part of the concept of cooperative knowledge. Note that, the warnings and vehicle take-over should be limited to situations evaluated with a high degree of certainty and personalization. The redundant information in terms of sources, modules, and communication paths, as well as the prior and continuous characterization of drivers and their driving styles, prevent the unnecessary modification of driving styles and regulates those aggressive behaviors that represent a danger.

### 5.1. Actual State

The discussion of this section begins by grouping the works that use the devices that allow the driver to operate on the vehicle while driving to capture the information of the driver–vehicle interaction, obtained through the internal communication buses of the vehicle, for example, the CAN bus. Although this information is present in most modern vehicles, it is not currently sufficiently exploited by the automotive industry to establish a personalized interaction between vehicle and driver. Wang et al. [101] propose a method to predict the driver’s braking intention in car-following scenarios from a perception–decision–action perspective according to their driving history, considering the following variables: vehicle speed, distance between the host vehicle and vehicle preceding, and the relative speed between them and TTC. The system combines a Gaussian mixture model (GMM) with a hidden Markov model (HMM) to infer the driver’s braking action given the state of the driving situation, using data from the CAN bus, cameras, and radars installed in the vehicle. Dang et al. [102] design an adaptive cruise control system with a lane change assistant. First, the risk associated with the lane change is analyzed by calculating the minimum safe space between the host vehicle and the surrounding vehicles. To calculate this risk, a driving style factor is introduced, which modifies the calculation of the minimum safety distances between the vehicle performing the maneuver and neighboring vehicles. The value of this factor is set by the driver. Values greater than one indicate conservative conductors, lower values indicate aggressive conductors. Finally, a coordinated control algorithm is developed using a predictive model control theory that limits the longitudinal acceleration of the vehicle to guarantee a better performance in terms of comfort in movement.

Zhu et al. [103] propose a personalized driver assistance system that includes driver profile identification for lane change assistance. Initially, the data obtained through driving simulators are analyzed and statistically processed to select the most relevant variables. With these results, the fuzzy c-means clustering algorithm is used to extract different conduction profiles. Three clusters are generated, which are associated with aggressive, normal, and conservative profiles. A neural network classifier, optimized by a particle swarm algorithm, is used to detect these conduction profiles. According to the profile of the driver identified by the system, preset values are determined for the execution time of the lane change maneuver and the minimum safety distance margin required with the preceding vehicle in the lane to which the change is made to avoid a forward collision. These personalized values are included in the analytical model to calculate the risk of collision associated with lane change and the subsequent behavior of the following vehicle in the destination lane. Su et al. [104] design a forward collision system that employs a method to recognize driver intent and driving behavior, based on the GMM. The proposed system has the advantages of adapting the model and the ability to generate probability densities in arbitrary shapes. In addition, it allows real-time implementation and has high precision. A precise recognition model of the driver’s driving behavior is first established and verified in a real-time driving simulator with 36 drivers as samples. Then, an FCW algorithm is designed with a braking execution strategy and an alarm classification based on the results of the driver’s driving behavior recognition. Drivers are classified as conservative, normal, and reckless. Mantouka et al. [105] identify driving styles based on unsupervised classification techniques, using acceleration and speed data collected through the use of smartphones. Initially, the styles are grouped into aggressive and non-aggressive behaviors to subsequently analyze additional unsafe behaviors associated with distraction and risk taking. Once the driving profiles have been detected, the driver’s average behavior and its persistence or volatility in different situations are analyzed.

Works that use the vehicle driving devices to capture the information of the driver–vehicle interaction have been grouped here, including the use of devices that emit visual, sound, or vibro-tactile alerts to the driver and analyze their reaction to these alerts. Although the use of devices that alert the driver is increasingly being addressed by the automotive industry in modern vehicles, the personalized interaction between vehicle and driver, which also includes the issuance of personalized warnings according to the driver’s behavior, continues to be a challenge in this area. Yang et al. [106] propose a collision warning system based on V2V communication. The algorithm initially detects the intention of the driver of the preceding vehicle, which is transmitted, together with other movement parameters of said vehicle, to the following vehicle through the V2V communication module. Finally, the safety application that runs in the following vehicle estimates the risk of potential collision with the information received through V2V and with the movement parameters of the vehicle itself, to alert the driver, outperforming systems based on TTC thresholds. Bavendiek et al. [107] present a human–machine interface design method based on the concept of metaphors to analyze and improve the vehicle–driver interaction through the design of HMI interfaces. This methodology, whose best known example is the design of computing machines based on the desktop interface, enables a friendlier relationship between man and machine, achieving improvements in the design of the HMI interface installed in the vehicle. The study focuses on describing a procedure to identify metaphors in the HMI environment of the automobile. Subsequently, the development of new HMI concepts based on the identified metaphors is proposed.

Iranmanesh et al. [108] design a FCW system that considers the driver’s historical braking profile in the face of previous alert events to establish a reaction threshold. The time of advance (time headway) for a vehicle is understood as the time it would take for the vehicle to travel circulating at its current speed and the distance between its front part and the front part of the vehicle that precedes it. The main metric is the alert activation, which is defined as a warning-triggering threshold, considered as the normal level of risk tolerance, to determine the need to issue an alert to the driver in the event of a possible dangerous situation. The caution–deceleration threshold is also defined to avoid false alerts and discard situations outside a dangerous situation, such as stops or turns in the presence of traffic lights. The system continuously monitors driver distraction through data obtained from the CAN bus associated with the state of the acceleration pedal, speed, acceleration, and turning angle, among others. The detection of driver distraction is performed by SVM classifiers and a multi-layered perceptron neural network. Sun et al. [109] propose a lane change assistance system based on the identification of the individual driving profile, determining an optimal alert threshold that varies as the characteristics of the individual profile change. The authors use the signal detection theory (SDT) to develop a method to determine the characteristics of the driver in a lane change maneuver. The target signal is defined as the operation to complete the maneuver by the driver and the noise signal is the operation to abandon the maneuver, which are associated with warning and non-warning criteria for the lane change warning system. The warning threshold is adjusted in real time, according to the particular characteristics of each driver during the maneuver. According to the authors’ definition, those who complete the lane change maneuver even in the presence of the safety system warning signal are grouped as aggressive drivers, and those who abandon the lane change execution even in the absence of the system alert signal are grouped as conservative drivers. Based on the analysis of the existing warning criteria, variables such as the TTC and the relative distance between the subject vehicle and the rear vehicle in the destination lane were used as warning indicators, while the initial warning threshold according to the difference of the speed of the subject vehicle was selected for the adaptive algorithm.

Choi et al. [110] propose a personalized design of the next generation HMI interfaces, where the driver can personalize the way in which he/she interacts with the vehicle and, in turn, it responds in a personalized way, identifying the characteristics and the state of the driver in a given situation. The proposed system consists of elements such as sensors embedded in the car, an adaptive inference engine that analyzes the driver–vehicle interaction, and an advanced digital platform in the vehicle cabin, which accesses the data obtained from this interaction. Dargahi Nobari et al. [111] propose a control scheme with feedback that considers the state of the driver as an input element for the system that analyzes the driver–vehicle interaction. Sensors (e.g., eye-tracker, physiological sensors) are used to detect the state of the driver. This result is compared with previously established situations to then design a policy that regulates the generation of stimuli tending to reduce the degree of criticality of the traffic situation. Table 4 summarizes the main works consulted that address elements of the driver–vehicle interaction. The directionality in which this interaction is approached is analyzed, as well as the way in which it is implemented.

### 5.2. Current Challenges

The current challenges of this section are focused mainly on the personalization of the ADAS and the motion modeling.

(i) The personalization of the ADAS, which should consider the driving profile, preferences, and peculiarities of the driver. In the design of HMI devices, not only the driver’s action on elements of the vehicle that determine its state of movement should be considered, but also feedback elements through visual or sound information that alert the driver about the consequences of their actions and that of the other actors in the environment regarding road safety. There is a common assumption in the personalization of ADAS systems that the driver is more comfortably adjusted to systems that implement a driving style similar to their own, but in practice, determining that optimal driving style for each individual driver is a very challenging task. In current systems, the process of interaction between driver and vehicle through the HMI limits the driver’s ability to correct and shape the system to establish a driving style that provides greater comfort. This interactive exchange phase between vehicle and driver requires further development. Another aspect to deal with in greater depth is the fact that the personalization process must be conceived as a continuous process. It is not enough to limit the customization process to obtaining and establishing a personalized system, since the drivers, influenced by various internal and external factors present on the road, can modify their preferences and driving styles in certain situations. For this reason, this phase of interactive exchange between driver and vehicle must last over time as a continuous process, improving its usability characteristics and finally, its benefit for road safety.

(ii) Modeling the longitudinal and lateral motion is a challenging task, which includes predicting the driver’s intention in response to the dynamics of the surrounding environment. The accurate sensing of the surrounding environment and the prediction of the intent of neighboring vehicles represent the biggest challenges for modeling the lateral and longitudinal motion of a vehicle by considering human driving preferences in the process. When the distances between vehicles, pedestrians, and objects are small, the degree of precision of on-board sensors must be very high. Similarly, if we take into account that the movement of vehicles on the road is a complex scenario of interaction between several actors, it is crucial to have tools that allow us to infer the intention and future behavior of neighboring vehicles. The failure of any of these two elements can lead to a tragic result, the dynamic instability of the vehicle, and even a traffic accident. When dealing with a high risk of collision, the development of conservative algorithms is chosen, even if this sacrifices aspects of the system such as efficiency, comfort, and acceptance of drivers and passengers. Incorporating V2V and V2I communications into the system can overcome these limitations and inefficiencies of non-cooperative systems [112].

(iii) Vehicles with SAE level three automation allow the driver to freely participate in tasks not directly associated with driving, but the driver is required to be able to disconnect from these tasks to regain manual control whenever required by the system. This requirement, related to the term take-over, represents a challenge to consider in the study of D–V interaction. This capability demands that the driver carry out this process of returning to driving tasks, ensuring a smooth, and at the same time, safe transition towards taking control of the vehicle. This challenge requires a novel design when addressing the D–V interaction (between the driver and the automated system), which considers the driver’s ability to take control in real time [113]. Situational awareness may be diminished in highly automated driving environments compared to manual driving environments if drivers are engaged in non-driving tasks [114]. In [115], the authors present a novel predictive haptic take-over controller to further explore the safe and smooth interaction mechanism during the take-over of autonomous vehicles.

The works described do not show the required bidirectionality between driver and vehicle, nor a holistic integration between subsystems, which is essential for information redundancy and system reliability. However, the more specific challenges in D–V interaction are associated with the degree of customization of these HMI devices for the timely display of alerts to the driver, the high-precision modeling and prediction of the driver’s intentions and the execution of the maneuver, and the guarantee of safe take-over transitions. Note that these interaction challenges not only limit the implementation of the proposed C-ADAS but also the progress in autonomous driving.

## 6. Driver–Environment (D–E) Interaction

The analysis of bidirectionality in the study of the interaction between driver and environment D–E is relevant for the C-ADAS to be aware of how attentive drivers are to the environment and how they react to eventual risk situations. Let us analyze, for example, the information provided from the environment directed towards the driver, which can be a stop warning signal. This information can be received: (i) directly by the driver (E -> D), or (ii) indirectly, via a stop notification issued from an RSU and notified by the vehicle to the driver (E -> V -> D). Regardless of the way in which the driver perceives the stop order, the C-ADAS must guarantee the vehicle’s stop. In this sense, it is vital to determine as soon as possible if the driver is aware of their surroundings and will carry out the braking maneuver effectively. For this purpose, the C-ADAS must continuously monitor the driver behavior through sensors, cameras, and wearable devices on board the vehicle (D -> V). In this particular case, if the driver does not react adequately, then the C-ADAS itself can issue an alert to stop, contributing as a redundancy path that guarantees that the information that the driver should receive is received satisfactorily (D -> E, D -> E), but if the risk situation cannot be fully mitigated by warning or vehicle take-over, then a general stop warning should be issued to alert other drivers and vehicles of the situation (D -> V -> E). It is important to take this redundancy route into consideration by the C-ADAS, given that there may be innumerable occasions in which the driver fails to perceive relevant information in terms of road safety, either due to distraction or due to impairments in their driving operational capacity. This degree of redundancy can only be achieved if the design of the C-ADAS is approached from a holistic and systemic perspective, where the study of bidirectionality in interactions is considered. The works of the state of the art, which are described in this section, fundamentally contribute to the indirect ways of obtaining this information and, in this sense, they are relevant in terms of the use of sensors, cameras, and wearable devices to monitor the status and behavior of the driver.

The importance of the natural perception’s module lies in the fact that the drivers form a personal reality about their surrounding environment, and make their own assessment of risk situations and the interaction between the different actors on the road, and based on this, the module determines their primary behavior before interacting with the ADAS. This module of natural perception has an amazing capacity to process large volumes of very diverse data, and this versatility of processing in humans still represents a challenge for the computer systems that exist in the present. However, it also has real limitations regarding the level of precision when estimating the kinematic variables associated with the movement of vehicles, something that is vitally useful in potential-risk situations on the road. This can be seen reflected in the deficient calculation of the adequate distances to carry out lane change or overtaking maneuvers, in maintaining an adequate safety distance with the vehicle in front, and detecting the presence of vehicles approaching from behind in adjacent lanes, among other common situations that arise daily. As an active element within the dynamics of the road environment, the drivers modify the state of said environment through their actions and reactions. The use of various signals by the drivers represents how they interact with the surrounding environment. Among these, we can mention: (i) visual signals through the vehicle’s lighting devices, such as turn signals, position indicator lights, or the stop lights; (ii) sound signals through the use of the horn; and (iii) body signals through the gestures or so-called “hand signals” of the drivers. The technological development associated with the infotainment area, together with the challenges imposed by the increase in the complexity of the road infrastructure and the increase in traffic congestion, have shown a growing use by drivers of navigation applications through the use of maps for route planning, which to some extent conditions their future behavior and mobility.

### 6.1. Actual State

The discussion of this section begins by grouping the works that use cameras to capture the information of the driver–environment interaction, a method that is currently widely used in the automotive industry to analyze the behavior of the driver inside the vehicle. Qiao et al. [116] propose a fatigue detection system using images of the driver’s face, eyes, and mouth, obtained by a smartphone camera. Signals of fatigue such as the blinking of the eyes are detected by means of a Haar qualifier [117], while sudden movements of the head are captured by calculating the variance of the centroid of the face. Yawning is detected by measuring changes in the geometry of the mouth through the Canny active contour method [118]. Mandal et al. [119] propose a fatigue detection system for bus drivers, using a percentage of eyelid closure (PERCLOS) method to determine the level of eye opening. Initially the system locates the position of the driver’s head in the incoming image to detect the location and orientation of the eyes. Yuen et al. [120] propose a system to monitor the driver’s activity during driving, analyzing information related to facial reference points, which is used for the detection of the face and the position of the head. Its performance is analyzed under various lighting conditions and degrees of occlusion of the driver’s face, which enables the system to be able to detect when there are occluded parts of the face and consequently achieve better estimation results in this situation.

Next, the works that use smartphones to capture the information of the driver–environment interaction are grouped together, a method that allows, in addition to the use of the smartphone camera, the use of the various sensors embedded in it, which allow for an obtaining of various physical variables that favor a more complete analysis of the driver’s actions while driving. One of the principal causes of vehicle accidents is distraction during the driving process. Eraqi et al. [121] present a vision-based system that uses RGB images obtained from the rear camera of a smartphone to recognize distracted driving postures, composed of a face detector, a hand detector, and a skin segmenter. The proposal is implemented using convolutional neural networks (CNN), obtaining results of the order of 90% accuracy; however, its performance overhead is higher in a real-time configuration. Janveja et al. [122] present a smartphone-based system for the detection of driver distraction by analyzing gaze tracking through the left and right rear-view mirrors and for fatigue detection by monitoring yawning and eye-blinking frequency. The system is designed to operate in low-light conditions using two configurations: in the first, a near-infrared (NIR) LED coupled to a smartphone is used; in the second, a generative adversarial network is used to synthesize a thermal image obtained from the RGB camera of a smartphone. The results show a better behavior of the system when NIR images are used. Kapoor et al. [123] designed a smartphone-based driver distraction detection system capable of operating in real time, which alerts the driver with a beep once distracted behavior is detected. The 10 classes of distracted behavior are drawn from the State Farm Distracted Driving Database [124], which is used for fine-tuning the four pre-trained CNN models, namely MobileNetV1, InceptionV3, VGG-16, and MobileNetV2. Xie et al. [125] present a system for the detection of driver distraction based on the use of data from GPS and the IMU sensors of a smartphone during the performance of turning maneuvers, lane change, lane maintenance, stops, and near stops. According to the results published by the authors, the best performance of the distraction detector, in terms of the F1 score metric, is obtained for lateral maneuvers (turn and lane change). This is assumed to be because the data used are more sensitive to this type of movement. This metric is defined by the authors themselves, using sliding windows to extract the temporal characteristics of the data obtained from the sensors and weighting the classification result of each sliding window according to the number of normal driving or distracted driving labels.

Next, we describe a work that uses the devices and internal sensors of the vehicle, which are grouped to capture the information of the driver–environment interaction, through physical variables that they describe as the result of the driver–environment interaction directly in the vehicle. Hu et al. [126] present a system to detect abnormal driving behaviors, such as recklessness, fatigue/drunkenness, and use of a smartphone. Unlike other works, where the driver’s activity is monitored by cameras that capture the image and movements of the driver while driving, in this study the authors analyze vehicle movement patterns, such as sudden acceleration and braking with a delayed response to traffic conditions, to detect abnormal driving behavior. To quantitatively assess these behaviors, a driver abnormality index is proposed. Qi et al. [127] present a passenger and driver activity detection system by means of acoustic recording devices for the recording and inference of activity inside the vehicle and by means of IMU and GPS sensors, including OBD-II [128] data for human activity detection and for the detection of vehicle movement patterns, such as braking, lane changes, and turns.

The works that use wearable devices and biomedic sensors have been grouped to capture the information of the driver–environment interaction, which allows a deeper analysis from the biological and physical point of view of the behavior of the driver during driving. The use of these devices, however, presents as a challenge the design of less invasive systems for the driver, with the aim of not distracting them or making them uncomfortable during the driving process, together with the necessary medical restrictions so as not to compromise the driver’s health. Rohit et al. [129] exploit the use of wearable EEG sensors for the real-time detection of driver drowsiness. An SVM classifier is used to detect drowsy states, by means of a spectral analysis of the EEG signals obtained from the drivers. Additionally, the blink duration parameters are extracted and analyzed, which were less favorable than the spectral analysis for the detection of drowsiness. Li et al. [130] develop a system for the early detection of driver drowsiness using not only the signal obtained from a wearable EEG sensor, but also by incorporating a gyroscope coupled to the driver’s head to analyze the movement of the driver’s head. Different from previous works, this study analyzes the feedback of the system when stimulating the driver by means of transcranial direct current to improve their state of alertness in real time while driving. In the same way, visual and vibro-tactile alerts are presented to the driver to combat the different levels of drowsiness detected by the system. Guo et al. [131] present a study to analyze the transition process of the driver’s intention, caused by the driver’s emotions. Various visual, olfactory, and auditory stimuli are used to generate emotions in the driver before driving tests and maintain them during the driving process. The results show a high accuracy and reliability to estimate the driver’s intention through the evolution of their emotions, and this system can be used to design personalized driving alerts in the human–machine interfaces of modern vehicles.

Finally, works have been grouped in which the driver plans their future route through the selection of points of origin and destination, which may undergo modifications according to the dynamic characteristics of the environment. Through the ADAS assisted by navigation maps, the driver is an active agent that modifies the environment, and through route planning, the state of the road environment is modeled. An assistance opportunity could be designed based on specific sections present in the routes defined by the driver, considering traffic between roads with different vehicle-flow capacities, entering roundabouts, overpasses, joining or exiting motorways, lane changes, and turns at intersections, among others. All these maneuvers involve risk, and through a cooperative knowledge of the pre-established routes, this risk can be minimized at these critical points for road safety. In these cases, an explicit notification prior to carrying out maneuvers at these points could mean the difference between the occurrence or not of a road accident. Xu et al. [132] present a study focused on improving intelligent transport management to combat the problem of predicting road congestion levels in real time. To do this, they develop an analysis method implemented in big data and cloud-computing platforms, which enriches the traditional method based only on the use of historical driving data, while also incorporating the users’ travel plans contained in the vehicle navigation information, which is associated with route planning. The system visualizes the data from this analysis by means of heat maps and sends personalized notifications to drivers according to the particular situation of their road environment.

Withanage et al. [133] develop a personal navigation system that simplifies the user’s interpretation of the translation of voice commands and the visualization of the routes in the navigator. During this process, the system initially translates the audio files into text using automatic speech recognition (ASR), and then uses natural language processing (NLP) techniques to retrieve previously undetected navigation information, finally displaying the generation of trajectories on the map using the development interface provided by Google Maps. Keerthana et al. [134] designed a navigation assistant based on voice instructions as a human–machine interface to guide the user to the requested destination through text-to-speech techniques to show the source and destination addresses on the map, allowing the planning algorithms to obtain route information by recognizing the user’s voice instructions. The addition of voice recognition techniques represents an improvement to the navigation tools for an individual client who needs to navigate in an obscure landscape. Zhou et al. [135] propose a system for planning tourist routes by correlating data associated with tourist places with the precise data of personal interest of the individual. To do this, it studies the behavior and personal needs of tourists and, based on this, it proposes tourist places with views and characteristics that are related to the interests previously analyzed. This method allows to obtain a more personalized route planning and its viability is testified through the design and execution of experiments with real data. Rathnayake et al. [136] present an interactive system for planning and evaluating travel routes, which analyzes the distances and weather conditions of the moment to carry out an evaluation of the trip previously established by the user. As a result of this analysis, recommendations and improvements to the initial travel plan are established, guaranteeing the optimization of the journey in terms of the distance and coverage of the different destinations selected a priori by the user.

Here, we consider driving behavior by capturing and detecting from on-board sensors of the bodily and physiological reactions that the driver manifests regarding stimuli from the environment. The knowledge of the D–E interaction can be correlated with the knowledge of the D–V interaction to be able to conclude about the ability or not of the driver to respond to certain events. Through V2V communication, the C-ADAS knows that the ego vehicle is approaching the vehicle in front of it (E–V interaction), simultaneously, through on-board cameras, it perceives drowsiness in the driver (D–E interaction) and, in turn, low pressure on the brake (D–V interaction). Consequently, the intervention of the C-ADAS in driving is required to avoid the accident, given the inability of the driver to react appropriately. Table 5 summarizes the main works consulted that address elements of the driver–environment interaction. The directionality in which this interaction is approached is analyzed, as well as the way in which it is implemented.

### 6.2. Current Challenges

The current challenges of this section are focused mainly on capturing the information associated with the driver’s natural perception process and to minimize the degree of distraction to the driver.

(i) Determining the main implicit and explicit characteristics that allow a capturing of the information associated with the driver’s natural perception process about the road environment. Actually, the most used explicit characteristics to describe the driver’s status are the movement of the eyes, head, hands, and feet, which reflect the state of attention of this and the body’s actions of reaction to the traffic’s dynamic. These are mostly monitored through vision and infrared cameras installed inside the vehicle, which are susceptible, in many cases, to adverse environmental conditions and poor visibility conditions.

(ii) How to make the sensing of these characteristics as least invasive as possible in order to minimize the degree of distraction to the driver. On the other hand, the implicit characteristics are related to drunkenness, drowsiness, and blood pressure or heart rate, among others, which can be obtained through various types of biomedical sensors attached to the driver. In this case, the installation of these devices can sometimes be inflexible in the face of driving dynamics and even a possible element of distraction and discomfort for the driver himself.

(iii) To address in greater detail the study of the influence that the driver exerts during the driving process on the road environment, an aspect that is evidenced in Table 5 with the reduced number of works that consider the directional interaction D -> E.

(iv) The integrated and complementary approach for the natural context perception module. In this process, the obvious physical limitations of the driver must be considered to analyze aspects related to the calculation of distances and relative speeds between the different vehicles, pedestrians, and objects on the road. It is necessary to establish a holistic approach to develop a comprehensive surrounding environment model from the vision through the sensors and the vision through the driver. This integration to analyze the driving behaviors in the dynamics of the environment will allow a significant increase in the prediction horizon and the precision in the prediction of the driver’s intention.

Studies related to the D–E interaction have addressed to a very low degree the correlation between the interpretation of the environment by the driver and the notification to the environment of this, which would undoubtedly modify the situation of the environment itself. The proposed C-ADAS architecture could take advantage of situations of vehicular congestion, unfavorable weather conditions, or dangerous situations of an effective bidirectional interaction between the driver and the environment, which would moderate the behavior of all drivers and consequently mitigate the risks in the environment. In addition to this, there are also the challenges associated with the use of these wearable devices and other on-board sensors and cameras and the way in which they are implemented, so as not to generate distraction or discomfort in the driver, while being able to obtain diverse information with a high predictive value in terms of inferring its future actions. The current state of progress made in the interaction between the driver and the environment, as well as the latent technological challenges, still limit the potential of the proposed C-ADAS.

## 7. System Evaluation

This section presents aspects related to the evaluation of C-ADAS in the field of road safety, the main mechanisms used, and the evaluation environments, as well as the main evaluation metrics used. We also include some of the challenges associated with the limitations in the use of simulators and evaluation metrics employed in order to meet the design requirements of a C-ADAS proposal, such as the one presented.

### 7.1. Evaluation Mechanisms

Road traffic represents a complex system made up of multiple independent and interrelated elements. In terms of safety, economy, and fluidity, the evaluation of transport can be associated with the performance of certain main factors, which can be grouped into elements such as the driver, the vehicle, and the road environment. For the most part, the interrelationship between these elements is highly variable and random in nature. The behavior of transport, from the mathematical point of view, can be associated with a stochastic process, where each of its elements can be represented as random variables. An in-depth study of the complexity of this system requires tools that allow for the achieving of a certain degree of reproducibility of the real characteristics present on the roads. To this end, driving simulators have been developed, which are devices used to simulate the driving of a vehicle in an environment with conditions similar to the real characteristics of road traffic. These devices are an effective tool in the training and study of driver behavior, but they also play a very important role in the design and improvement of the HMI elements present in vehicles.

Among the many advantages of using these devices, we can mention the possibility of studying the driver’s behavior in situations that may arise while driving and even emulate this behavior in situations that are legally prohibited, such as when driving under the influence of alcohol, or using a cell phone while driving. Another advantage of using simulators corresponds to the reduction of the economic costs associated with carrying out tests in real environments, allowing the generation of high volumes of data from very diverse situations for the training of artificial intelligence models, which guarantees the conditions of reproducibility necessary for statistical analysis of these situations. However, the main limitation is that in these scenarios, drivers are not exposed to a fundamental element: the real risks present on the roads. This can distort the analysis and modeling of the behavior of drivers in real situations. In the same way, other elements of distraction and influence that are only present in real scenarios are not considered within the simulators, which shape the behaviors of drivers and their reactions to risky situations.

Musa et al. [137] point out that vehicle manufacturers develop and test ADAS technologies following the V-model, which is conventionally used for automotive electronics. The development steps consider the model-in-the-loop and software-in-the-loop tests, while in the validation, the hardware-in-the-loop, vehicle-in-the-loop, and driving tests can be performed. In addition, model predictive control (MPC)-based strategies represent interesting solutions for ADAS evaluation. Vehicle-in-the-loop tests may be useful to evaluate the effective responses of vehicles in safety-critical scenarios and to test it with real-V2X systems, while the combination between hardware-in-the-loop and driving simulators is useful for taking into account human error.

### 7.2. Performance Metrics

The overall performance of a C-ADAS system must consider not only the precision in estimating the location of remote vehicles, but also the impact of the communication schemes used and the performance of the communication network. The first step in determining this performance is to define a metric that can directly indicate how the security algorithm is performing quantitatively. Generally, for the evaluation of this type of situation, classification metrics are used. Table 6 summarizes the most used metrics in the review of the literature consulted, grouping them fundamentally in regression, classification, and communication performance metrics. The accuracy of hazard detection algorithms to classify situations as dangerous is one such metric [138]. The security algorithm generally runs periodically and at each run-time instance it determines whether a threat exists. The ratio between true positives plus true negatives within the total number of execution instances of the algorithm is defined as the precision. Position-tracking error (PTE), which describes the mean or 95th percentile of the error in tracking a remote vehicle’s position, is used to ensure accuracy in position estimation, which is an independent metric of the C-ADAS that is running. Warning detection is performed at the same time as position tracking. The overall performance of C-ADAS depends on the accuracy of the alerts, which depends on the accuracy of tracking the position of the remote vehicles, and this, in turn, depends on the performance of the communications. To analyze this performance, metrics such as the packet error rate (PER) are used, which is the number of packets received incorrectly divided by the total number of packets received. Another metric that can be used to model the robustness of the communications network is the packet loss ratio (PLR), equal to the number of packets not received divided by the total number of packets sent. In this sense, we analyze the evaluation metrics employed in the surveyed works, grouping them into: (i) regression metrics, (ii) classification metrics, and (iii) communication performance metrics. The fulfillment of this task requires the use of evaluation metrics that consider all the elements of the system, focused on the requirements associated with RSA, described mainly in terms of road safety. The robustness of the information is referred to as guaranteeing the reliable delivery of the road safety information required by the RSA (low rates of losses and transmission errors). The freshness of the information meets the delay requirements to guarantee the usefulness of the same in terms of road safety.

Through the analysis shown in Table 6, we can observe as a flaw present in most of the analyzed works, that the consideration, and even more so the evaluation of the behavior of the communications, has not been sufficiently addressed. We assume that this is due to the fact that currently there is a majority trend in the automotive industry to bet on the use of different sensor technologies in vehicles, which are capable of self-supporting the development of the ADAS implemented in these without considering the cooperative operation of these systems between different vehicles and/or with the road infrastructure.

### 7.3. Current Challenges

An important aspect in the evaluation of the C-ADAS systems is the fact of considering their operation and degree of customization as a continuous process. In this sense, systems designed to operate in real time must be capable to adjust dynamically with the new real data. Together with this, it is also important to consider the use of real databases obtained in field measurement campaigns to provide the system with a priori knowledge, which can be complemented with the processing of data obtained in real time during the tests of system operation. Currently, there is a lack of development of integral evaluation systems in order to be able to evaluate our proposal. Note that, it is impossible for a current simulator to fully replicate real-world driving scenarios, especially from a vehicle and traffic perspective, since there are a set of events and random variables that are difficult to model, such as catastrophic, climatic, environmental, and political–social–cultural events, however, these events have a high incidence on vehicular traffic, the environment, and drivers. Moreover, several of the interactions should be evaluated beforehand, such as the interaction of vehicles to each other as a result of collaborative actions and the communication delays between them. Besides, we consider that it is essential for the effective implementation of the proposed C-ADAS to run as many tests as possible in controlled scenarios, but to make them as real as possible. In the short or medium term, these tests must be executed in hybrid environments where there are vehicles with sensing and communication capabilities and others with the absence of some or none of these capabilities. The latter shows the current relevance of the driving assistance systems, even though there is a growing development in the area of autonomous driving.

## 8. Conclusions

In this survey, we describe the main points related to the design of a C-ADAS from a holistic and systemic perspective. It is reinforced that the premise of a C-ADAS must take into consideration the three fundamental elements described, namely the driver, vehicle, and environment, and must be designed on an abstraction layer higher than the interaction plane of these elements, which supports the principal module of the C-ADAS system. The main challenges that this industry must set itself must be focused on understanding the current challenges present in each of the three areas of interaction described throughout this work:From the point of view of vehicle–environment interaction, we can highlight the standardized incorporation of sensing and communication devices in new vehicles, the reduction in the cost of sensor technologies, and the development of the necessary infrastructure to support the networks of vehicular communication;In relation to the vehicle–driver interaction, work should be done to achieve an increase in the degree of customization of the ADAS system, considering the particularities with which each driver acts on the vehicle’s driving elements, as well as the way in which it reacts individually to the notices and alerts issued by the system;Work must also be done on the design of a flexible HMI interface that adapts continuously to the behavior of the driver in various situations that arise on the road, which is capable of switching between the driver assistance function or taking control of the vehicle before a situation of danger that the driver is not able to resolve favorably;One of the areas where greater action is required is in relation to the driver–environment interaction, and the detection of the driver’s intention remains today a latent challenge in the design of ADAS systems. To do this, the study of the explicit and implicit characteristics of the driver’s behavior must be promoted through the use of devices attached to the interior of the vehicle, which must act in a minimally invasive way so as not to distract or cause general annoyance to the driver, while being able to capture the greatest amount of relevant information from them.

We consider that the necessary future work is the design of a C-ADAS oriented to a higher level of cooperative systems, where cooperation goes beyond the exchange of information provided by the use of communications. We refer to the fact of considering the risk-estimation and decision-making process as a consensual act between different road users. In this approach, the C-ADAS of a vehicle includes, as an additional input in its decision making, the results of the decisions of C-ADAS implemented in neighboring vehicles. The concept of the IoV, a cloud-based service, and the emerging technologies such as the fifth generation of mobile communications can play a crucial role in the sensing and exchange of the information associated with the three main interactions described in this work, namely driver–vehicle, driver–environment, and vehicle–environment, managing the integration of many data sources, which are sometimes very different from each other. 

## Figures and Tables

**Figure 1 sensors-22-03040-f001:**
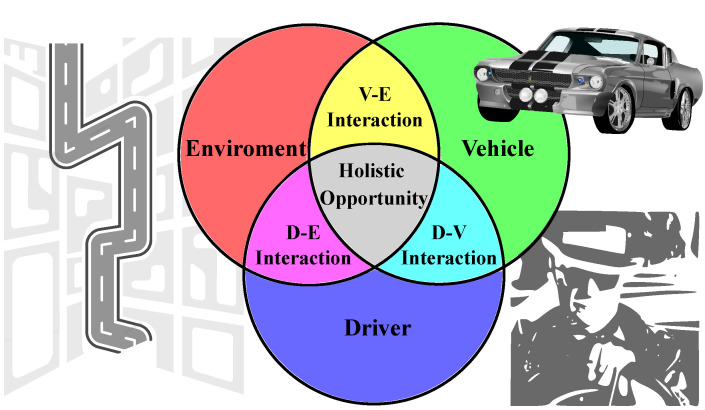
Main elements involved in a road safety system: environment, vehicle, and driver. Interactions between these elements: vehicle–environment (V–E) interaction, driver–environment (D–E) interaction and driver–vehicle (D–V) interaction. At the center is reflected the holistic relationship between elements and the opportunity in the C-ADAS design.

**Figure 2 sensors-22-03040-f002:**
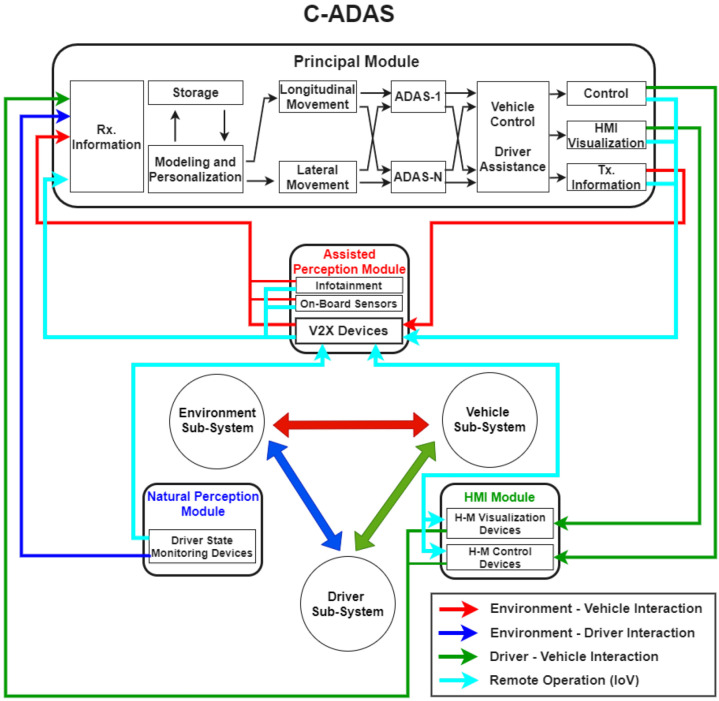
Conceptual architecture of a cooperative advanced driver assistance system (C-ADAS) from the holistic and systemic vision. The red, green, and blue lines indicate the local operation of the principal module (inside the vehicle), while the cyan lines refer to the remote operation of the principal module through the vehicular network under the IoV concept.

**Table 1 sensors-22-03040-t001:** Analysis of the main characteristics and limitations of other published surveys and this work in terms of architecture, system evaluation, and interactions between elements, considering D–V interaction, D–E interaction, and V–E interaction; specifying whether this relationship is bidirectional (<->) or unidirectional (->). Cases where these topics are not addressed (N/A) are also highlighted.

Articles	Architecture	D–V Interaction	D–E Interaction	V–E Interaction	System Evaluation
[18]	N/A	D <-> V	N/A	V <-> E	N/A
[19]	Modular	D <-> V	N/A	N/A	Mechanisms
[20]	N/A	D <-> V	N/A	E -> V	N/A
[21]	Modular and relational	D <-> V	D -> E, E -> D	E -> V	Mechanisms and metrics
[22]	N/A	N/A	D -> E, E -> D	V <-> E	N/A
[23,24]	N/A	N/A	N/A	V <-> E	N/A
[25]	N/A	N/A	N/A	E -> V	N/A
[26]	N/A	N/A	D -> E, E -> D	N/A	N/A
[27,28]	N/A	D <-> V	D -> E, E -> D	N/A	N/A
[29]	Modular	D <-> V	N/A	V <-> E	N/A
This work	Modular, holistic, and systemic	D <-> V	D <-> E	V <-> E	Mechanisms and metrics

**Table 2 sensors-22-03040-t002:** Analysis of the ADAS architecture proposed in the revised surveys and the architecture proposed in this work.

Articles	Control System	Modular Structure	Personalization	Cooperative Communication	Assistance Function
[19]	Local control	System composed of three main modules: ADAS module, personalization module, and HMI module.	Personalization module that continuously adapts the ADAS to the driver’s behavior through the HMI interface.	Cooperative communication is not explicitly considered.	Consider vehicle control and multiple driver assistance functions.
[21]	Local control	System composed of six main modules: environment perception module, vehicle dynamic module, driver behavior recognition module, driver intention inference module, lane change decision module and interaction module.	The interaction module models the driver hand and foot dynamics as well as the dynamics of the vehicle–control interface.	Cooperative communication is not explicitly considered.	Consider vehicle control and a single assistance function: lane change intention inference.
[29]	Local and remote control	System composed of three main elements: sensing and communication technologies, human factors, and information-aware autonomous vehicles controllers.	Human factor element: design of a CAV system based on human driver’s expectation, and adaptation of the human driver to the designed CAV system.	It considers the exchange of environment information through the sensing and communication modules, but not the alerts and warnings generated for the ADAS system.	Consider vehicle control and multiple driver assistance functions.
This Work	Local and remote control	System composed of three modules that sense the interaction between the three elements of road safety: the natural perception module, assisted perception module, and HMI module. Besides, a principal module that receives, processes, stores, makes decisions, and transmits the safety information.	Continuously adapts the ADAS by monitoring the driver’s condition and their physical interactions with the vehicle’s driving-control elements, as well as its reaction to the information and alerts issued by the HMI module.	It considers the exchange of information through the communication devices on board the vehicle, both the information obtained by the three sensing modules and the alert or warning information generated by the main module of the system.	Consider vehicle control and multiple driver assistance functions.

**Table 3 sensors-22-03040-t003:** Works in which vehicle–environment interaction is addressed. The “ideal” behavior means that the authors do not consider the losses of the communication channel. The works that are not grouped within “ideal” behavior or “non-ideal” behavior, are those in which the use of communications is not considered for the design of ADAS.

Articles	Directionality		Implementation of the Interaction			Communications	
	**V -> E**	**E -> V**	**Data Source**	**Data Type**	**Environment**	**“Non-Ideal” Behavior**	**“Ideal” Behavior**
[82]		✓	Cameras, radar, LIDAR, and DSRC	Real	Real	✓	
[66]		✓	Radar	Simulated	Simulation		
[55,57,58,59,60,61,62]		✓	Cameras	Real	Real		
[63]		✓	Cameras	Real	Simulation		
[68]		✓	LIDAR	Real	Real		
[69]		✓	Radar	Real	Real		
[70]		✓	Cameras, radar, GPS, and IMU	Real	Real		
[83]	✓	✓	Cameras, LIDAR, and DSRC	Real	Real and simulation		✓
[71]	✓	✓	DSRC	Real and simulated	Simulation	✓	
[65]	✓		Cameras	Real	Real		
[84]	✓	✓	Cameras, radar, LIDAR, and DSRC	Real and simulated	Simulation		✓
[76,77,81]	✓	✓	DSRC	Simulated	Simulation	✓	
[78]	✓	✓	DSRC	Real and simulated	Real	✓	
[79]	✓		DSRC	Real	Real	✓	
[37]	✓	✓	DSRC	Real and simulated	Real and simulation	✓	
[86]		✓	Cameras, radar, and LIDAR	Real and simulated	Real and simulation		
[87]	✓	✓	DSRC	Simulated	Simulation	✓	✓
[88]	✓	✓	DSRC	Simulated	Simulation		✓
[89]	✓	✓	Cameras, radar, LIDAR, and DSRC	Real and simulated	Simulation		✓
[90]	✓	✓	Cameras, radar, LIDAR, and DSRC	Real and simulated	Simulation	✓	
[91]	✓	✓	DSRC	Simulated	Simulation	✓	
[92]	✓	✓	DSRC	Simulated	Simulation	✓	
[94]	✓	✓	DSRC	Simulated	Simulation	✓	
[93]	✓	✓	DSRC	Simulated	Simulation	✓	

**Table 4 sensors-22-03040-t004:** Works in which driver–vehicle interaction is addressed. The type of personalization described in the consulted works is analyzed from two approaches: (i) the action that the driver exerts on the driving elements of the vehicle (described as an action on the vehicle) and (ii) the reaction that the driver adopts before the notices and alerts sent by the system (described as reaction to ADAS).

Articles	Directionality		Implementation of the Interaction			Personalization Types	
	**V -> D**	**D -> V**	**Data Source**	**Data Type**	**Environment**	**Action on the Vehicle**	**Reaction to ADAS**
[101]		✓	CAN bus	Real	Real	✓	
[102]		✓	CAN bus virtual	Real	Simulation	✓	
[103,104]		✓	CAN bus virtual	Simulated	Simulation	✓	
[105]		✓	Smartphone sensors	Real	Real	✓	
[106,107]	✓	✓	CAN bus and warning indicator	Real and simulated	Real and simulation	✓	✓
[108,109,110]	✓	✓	CAN bus and warning indicator	Real	Real	✓	✓
[111]	✓	✓	Eye-tracker and warning indicator	Real	Simulation	✓	✓

**Table 5 sensors-22-03040-t005:** Works in which driver–environment interaction is addressed. In this work, we decided to analyze two of the characteristics most addressed in the literature consulted on detection of the driver state: (i) monitoring of driver fatigue or drowsiness (fatigue monitoring) and (ii) driver distraction. Furthermore, we analyze the works in which the driver has an active role in the modification of the environment’s state through the planning of their future route.

Articles	Directionality		Implementation of the Interaction			Driver Status	
	**E -> D**	**D -> E**	**Data Source**	**Data Type**	**Environment**	**Detection of the Driver State**	**Trajectory Planning**
[116,119]	✓		Camera	Real	Real	Fatigue monitoring	
[129]	✓		EEG and wearable sensors	Real	Simulation	Fatigue monitoring	
[130]	✓		Bluetooth sensor and smart watch	Real	Real	Fatigue monitoring	
[126]	✓		CAN bus	Simulated	Simulation	Fatigue monitoring, driver distraction	
[121,123]	✓		Smartphone camera	Real	Real	Driver distraction	
[122]	✓		Smartphone and NIR led	Real	Real	Fatigue monitoring, driver distraction	
[127]	✓		Microphone and CAN bus	Real	Real	Driver distraction	
[125]	✓		Smartphone sensors	Real	Real	Driver distraction	
[131]	✓		Visual, auditory, and olfactory stimuli	Real and simulated	Real and simulation	Driver distraction	
[120]	✓		Cameras	Real	Real	Driver distraction	
[132]		✓	Map display device	Real	Real		Route planning
[133]		✓	Smartphone	Real	Real		Route planning
[134]		✓	Smartphone	Real	Real		Tourism guidance
[135]		✓	Map display device	Real	Real		Tourism guidance
[136]		✓	Non specified	Real	Real		Tourism guidance

**Table 6 sensors-22-03040-t006:** Classification of the consulted works according to the evaluation metrics used.

Articles	Regression	Classification	Communication
[82]	90 percentile position error		Packet loss rate
[55,57,58,63]	RMSE position error		
[68]	Total euclidean error sum, horizon euclidean error, and modified Hausdorff distance [139]		
[69]	Mean absolute error		
[59]	Collision risk probability		
[60,70,104,108,116,121,123,131]		Accuracy	
[61]		Accuracy, true positive rate, and true negative rate	
[62]		Recall, accuracy, precision, and F1-score	
[83]	Absolute position error	Accuracy	
[71]	95 percentile of velocity and acceleration error	Accuracy	PER, transmission rate
[65]	95 percentile of absolute position error		
[84]	Spacing position error and velocity tracking error		
[76,77]	95 percent cutoff Euclidean position error		Transmission rate, percentage of packet losses
[78]	90 percentile position tracking error		Transmission rate, PER
[79]	90 percentile position tracking error, average model persistency		Packet length, transmission rate
[81]	Probability of safe breaking		Maximum communication delay, BER
[101]		Accuracy, sensitivity, and specificity	
[102]	Tracking position error		
[106]		Accuracy; premature, timely, and late warning rate	
[109]		Accuracy, false positive rate, and false negative rate	
[103]	Tracking position error	Accuracy	
[105]	Driver’s behavior volatility	Calinski–Harabasz, silhouette, and Dunn’s index	
[119]	Average error of eye-openness	Detection rate	
[129]		Precision, recall, and accuracy	
[126]	Abnormal driver behavior index		
[122]	Normalized mean squared error	Accuracy, precision, recall, and F1-score	
[127]		Accuracy, precision, and recall	
[125]		Precision, recall, and F1-score	
[120]	Mean and standard deviation of yaw angle absolute error	Detection rate, success rate	

## Data Availability

Not applicable.

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
