# Peer review of "Survey of Cooperative Advanced Driver Assistance Systems: From a Holistic and Systemic Vision"

_sensors, 2022, doi:10.3390/s22083040_

Round 1

Reviewer 1 Report

This work presents a survey of C-ADAS and describes a conceptual architecture that includes the driver, vehicle, and environment and their bidirectional interactions.

There are some comments that are listed as follows:

  1. The environment subsystem takes into account other vehicles, but the vehicle mentioned above is considered as a separate subsystem. It is recommended to bring the vehicle out of the environment and supplement a Vehicle-Vehicle interaction. (e.g., exclusively employ the use of vehicular communications to capture the information from the interaction with the vehicle. See: A Survey on Cooperative Longitudinal Motion Control of Multiple Connected and Automated Vehicles[J]. IEEE Intelligent Transportation Systems Magazine, 2020, 12 (1): 4-24.)
  2. The current challenges faced by Driver-Vehicle (D-V) Interaction need to be complemented. For example, realizing the takeover and handover of tasks, the transfer and handling of emergency tasks, and seamless transition between manual control and autonomous driving.
  3. There is only E->D or D->E in Table 5, while D<->E (the bidirectional interaction) is described in Table 1. There is no bidirectional interaction between the driver and the environment. Supplements can be made.
  4. The detection of drivers’ driving behavior and physiological state should be classified into autonomous Driver-Vehicle interaction.

Author Response

Dear Reviewer,

We would like to thank the Reviewer for handling critically and carefully our paper and for his/her suggestions, which have been very helpful to improve the quality of the manuscript. In the attached pdf file, we address each speciffic comment and highlight in blue, for your convenience, the changes made to the manuscript.

Sincerely,

The authors

Reviewer 2 Report

This survey delivers a holistic and systemic review of C-ADAS, where the bidirectional interaction among three fundamental elements (i.e., driver, vehicle, and environment) are specifically discussed. Correspondingly, relevant challenges in the development of C-ADAS and the evaluation methods are elaborated. Overall, the contribution of this paper is clearly stated, and the content is sounded. The reviewer has the following detailed comments that can help the authors to enhance the quality of this paper.

  1. It would be better to discuss the status of ADAS from some automakers with high market share (i.e., Toyota, GM, Ford, Volkswagen, etc.), and link the corresponding challenges and future trends to C-ADAS.
  2. The cooperative driving (e.g., CACC, intersection control) should be included in the discussion of V-E interaction, as the cooperative driving is the most significant outcome of sensing, communication, and automation technologies, which also significantly influences drivers’ behavior. The following papers might be helpful:
  • Cooperative method of traffic signal optimization and speed control of connected and automated vehicles, IEEE Transactions on Intelligent Transportation Systems 20 (4), 1390-1403.
  • Cooperative signal-free intersection control using virtual platooning and traffic flow regulation, Transportation Research Part C: Emerging Technologies 138, 103610.
  • A real-time deployable model predictive control-based cooperative platooning approach for connected and autonomous vehicles, Transportation Research Part B: Methodological 128, 271-301.
  • Smooth-Switching Control-Based Cooperative Adaptive Cruise Control by Considering Dynamic Information Flow Topology, Transportation Research Record 2674 (4), 444-458.
  • Smoothing traffic flow via control of autonomous vehicles, IEEE Internet of Things Journal 7 (5), 3882-3896.
  • A Survey on Cooperative Longitudinal Motion Control of Multiple Connected and Automated Vehicles, IEEE Intelligent Transportation Systems Magazine 12 (1), 4-24.
  • Review of learning-based longitudinal motion planning for autonomous vehicles: research gaps between self-driving and traffic congestion, Transportation research record 2676 (1), 324-341.
  1. In the D-V interaction, a very interesting topic is human-like autonomous driving, will C-ADAS stimulate the development and post new challenges on that? As the C-ADAS aims to help drivers to drive in a safer and more comfortable manner, will it induce human drivers to change their driving styles and potentially become less “human-like”?
  2. In the evaluation of C-ADAS, driving simulators cannot fully replicate real-world driving scenarios, especially from the vehicle and traffic perspective, which will inevitably generate biased data distribution. How would this disadvantage be tackled? Further discussion would make this paper more rigorous.

Author Response

Dear Reviewer,

We would like to thank the Reviewer for handling critically and carefully our paper and for his/her suggestions which have been very helpful to improve the quality of the manuscript. In the attached pdf file, we address each speciffic comment and highlight in blue the changes for your convenience.

Sincerely,
The authors

Reviewer 3 Report

In this work, the authors present a survey on cooperative ADAS from different interaction perspectives between a vehicle, its driver and its environment. 

The main issue with the paper lies with its structure as too many parts look more like a catalogue than a review. The authors should go beyond all the paper they have surveyed to provide more analysis on the key elements which can be taken.

For instance, chapter 2 starts with a long list of similar reviews which is then concluded that "none of them have described the ADAS architecture from a holistic and systemic perspective". This statement is not introduced nor justified. The reader cannot understand why having an analysis from such perspective is important. So, I would recommend to the authors to better integrate this part with section 2.1 which justifies why they believe this approach is necessary.

Chapter 3 gives the main concept which are going to be addressed by the paper. Then, chapter 4, 5 and 6 lacks clarity and some illustration. They provide long description of many works, but it is hard for the reader what should be taken from this and how it relates what is presented in chapter 3. Besides, many of the presented works are well established and could be presented in a shorter way as this brings few novelties.

As a conclusion, I would recommend to the authors to present in a more analytical way, and may be shorter, so that it would be straightforward for the reader to get the challenges which are to be highlighted.

Author Response

(The authors gave the same response as above.)

Reviewer 4 Report

The topic is interesting and of good level. For this reason. the following few modifications are requested:

-the novelty and originality of the work should be clearly stated

-Avoid lump sum references. In this regard, the introduction section should also refer to works on MPC (10.3390/en14237974)

-Conclusion section should provide main results in the form of bullet points for an easier reading.

Thanks

Author Response

Dear Reviewer,
We would like to thank the Reviewer for his/her suggestions which have been helpful to improve the quality of the manuscript. Unfortunately, we received the Editor's Revision noti cation on March 17 and your comments were not yet in the system (until March 19). We have become aware of your review, for the  first time, just at uploading the responses of the other Reviewers. However, without
time availability, we have responded to your comments, we hope that those are to your liking and that you are satis ed. In the attached pdf file, we address each speciffic comment and highlight in blue the changes for your convenience.
Sincerely,
The authors

Round 2

Reviewer 1 Report

The authors have addressed all my concerns.

Author Response

We would like to thank the Reviewer for handling critically and carefully our paper and for his/her suggestions, which have been very helpful to improve the quality of the manuscript.

Reviewer 2 Report

The authors have addressed all my comments. I encourage the authors to cite the following related papers

Optimal toll design problems under mixed traffic flow of human-driven vehicles and connected and autonomous vehicles

Multiclass information flow propagation control under vehicle-to-vehicle communication environments

Multiclass traffic assignment model for mixed traffic flow of human-driven vehicles and connected and autonomous vehicles

Optimal toll design problems under mixed traffic flow of human-driven vehicles and connected and autonomous vehicles

Author Response

We would like to thank the Reviewer for handling critically and carefully our paper and for his/her suggestions which have been very helpful to improve the quality of the manuscript. In the following, we address each specific comment and highlight in blue the changes for your convenience.

Reviewer 3 Report

The authors have added some content, especially to introduce and conclude most chapters which improves the paper and make it more comprehensive to the reader.

In the new added content, some wording could be improved as does not sound grammatically correct:

  • p.3 "The proposed C-ADAS so complete"
  • p.6 "which mainly seeks to appropriate the advantages that the IoV concept gives"
  • p.7 "CV2X" -> C-V2X
  • p.7 two consecutive sentences start with "Note that". It is quite heavy.

Author Response

We would like to thank the Reviewer for handling critically and carefully our paper and for his/her suggestions, which have been very helpful to improve the quality of the manuscript. In the following, we address each specific comment and highlight in blue the changes for your convenience.
